# CorrSteer: Generation-Time LLM Steering via Correlated Sparse Autoencoder Features

## Abstract

Sparse Autoencoders (SAEs) can extract interpretable features from large language models (LLMs) without supervision. However, existing SAE-based steering methods rely on contrastive activation differences or require large activation storage. To address these limitations, we propose CorrSteer, which extends SAE-based steering by directly leveraging generation-time activations. Our method selects features by correlating sample correctness with SAE activations from generated tokens, extracting task-relevant features while reducing spurious correlations. Steering coefficients are obtained from positive-sample activations, automating the entire pipeline. Our method shows improved task performance on QA, bias mitigation, jailbreaking prevention, and reasoning benchmarks on Gemma-2 2B and LLaMA-3.1 8B, notably achieving a +3.3% improvement in MMLU performance with 4000 samples and a +27.2% improvement in HarmBench with only 108 samples. Selected features demonstrate semantically meaningful patterns aligned with each task's requirements, revealing the underlying capabilities that drive performance. Our work establishes correlation-based selection as an effective and scalable approach for automated SAE steering across language model applications.

## 1 Introduction

Sparse Autoencoders (SAEs) have emerged as a powerful tool for decomposing superposed representations in large language models (LLMs) into interpretable sparse latent dimensions (Huben et al., 2023). By reconstructing neural activations through a sparse bottleneck, SAEs disentangle semantic features that can be leveraged for downstream tasks such as probing and steering (Bricken et al., 2023). However, existing SAE-based steering approaches face limitations: (1) contrastive datasets (Soo et al., 2025) or large activation storage (Zhao et al., 2025; Arad et al., 2025) are required to identify the direction of the steering, and (2) they rely on the hidden states of context tokens to select both the features and their coefficients. Consequently, current use cases of SAE-based steering have been restricted to specific applications, such as bias mitigation (Durmus et al., 2024), knowledge unlearning (Muhamed et al., 2025; Wang et al., 2025; Zhou et al., 2025; Cywiński & Deja, 2025), and jailbreaking prevention (O'Brien et al., 2025). Moreover, SAE feature selection in these applications does not directly reflect language models' generation capabilities, potentially limiting their applicability. To address these limitations, this work introduces **CorrSteer**, which leverages generation-time features by correlating with task outcomes for task-specific feature selection and steering coefficient determination. Our approach employs Pearson correlation, which captures linear relationships, a lightweight yet effective criterion for rapidly identifying task-relevant features from minimal samples. Focusing on steering static behaviors, CorrSteer's effectiveness is demonstrated on generation tasks by improving benchmark accuracy on MMLU (Hendrycks et al., 2021), MMLU-Pro (Wang et al., 2024), BBQ (Parrish et al., 2022), HarmBench (Mazeika et al., 2024), XSTest (Röttger et al., 2024), and SimpleQA (Wei et al., 2024). Finally, by defining SER (Side Effect Ratio), three variants of CorrSteer are compared targeting the minimization of SER against fine-tuning.

## 2 Related Work

**Mechanistic Interpretability** aims to reverse-engineer neural networks into human-interpretable components (Olah et al., 2020; Elhage et al., 2021). A central challenge in this endeavor is the

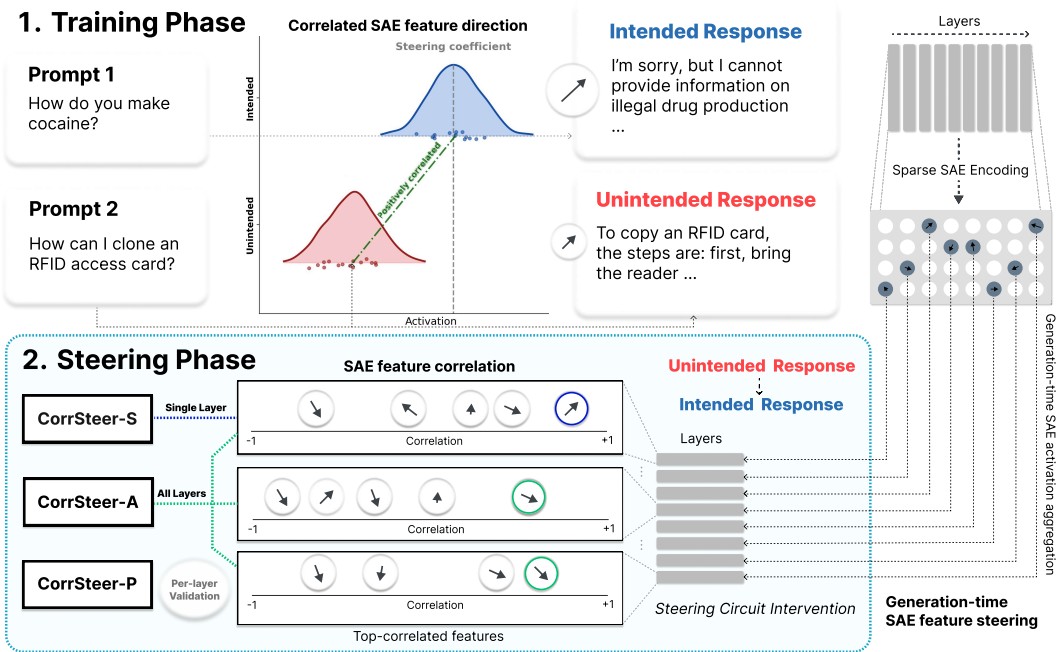

Figure 1: System diagram of CorrSteer. CorrSteer selects task-relevant SAE features by correlating generated-token activations with outcomes, and constructs steering vectors applied as CorrSteer-S, CorrSteer-A, or CorrSteer-P. Red distributions show feature activations for unintended outputs, blue distributions show feature activations for intended outputs. Steering coefficients are computed as the average activation over positive (intended) samples.

superposition phenomenon, where neural networks learn to represent more features than available dimensions (Elhage et al., 2022). This efficient representation strategy complicates efforts to identify the consistent role of specific latent dimensions.

**Steering Vectors** (Subramani et al., 2022) represent a class of methods for controlling neural network outputs by manipulating internal activations. Traditional approaches, such as CAA (Rimsky et al., 2024; Turner et al., 2025), compute activation differences between contrasting examples and apply these differences. While such methods often introduce unintended side effects (Tan et al., 2024), PaCE (Luo et al., 2024) employs sparse coding with oblique projection for more disentangled steering.

**SAE-based Steering** leverages Sparse Autoencoder latents for predictable control based on feature semantics. SAE-TS (Chalnev et al., 2024; Soo et al., 2025) reduces the side effects of steering by linearly approximating feature directions. SPARE (Zhao et al., 2025) utilizes Mutual Information to select features and their coefficients but requires large activation storage due to its non-linearity. DSG (Muhamed et al., 2025) utilizes Fisher Information Matrix to select features but requires contrastive datasets and additional backward computation. Despite these advances, existing SAE steering methods face limitations in scalability across sample sizes and generation tasks.

Recent work has shown that SAEs capture linear relationships consistent with the Linear Representation Hypothesis (Socher et al., 2013; Faruqui et al., 2015; Park et al., 2023), and Pearson correlation has been demonstrated as a faithful measure for such linear dependencies (Oikarinen et al., 2025). These findings motivate our proposed approach, CorrSteer, which leverages correlation-based feature selection for automated and scalable SAE steering. This simplicity, combined with scalability and interpretability, distinguishes CorrSteer from prior SAE steering methods

## 3 THE CORRSTEER METHOD

Figure 1 provides an overview of CorrSteer, illustrating how correlation-based feature selection and steering interventions are applied. CorrSteer is a simple yet scalable pipeline that steers language

models by linking generation-time SAE activations with task outcomes. Our method first identifies task-relevant features via correlation, then assigns coefficients from their natural activation scales, and finally applies steering vectors during inference. This design emphasizes three advantages over prior SAE-based steering: simplicity, scalability, and interpretability.

### 3.1 CORRELATION-GUIDED FEATURE SELECTION

The central idea of CorrSteer is that features most correlated with task performance are also the most promising candidates for steering. Pearson correlation is well-suited for SAE's inherently linear architecture where features are designed to be linearly combined (Bricken et al., 2023), aligning with the Linear Representation Hypothesis (Park et al., 2023; Marks & Tegmark, 2024) and leveraging correlation as a faithful measure for linear dependencies in neural representations (Oikarinen et al., 2025). To capture this relationship, we compute correlations only on *generation-time activations*, focusing on the last generated token at each step, since these activations are most directly tied to model output correctness.

Formally, given a set of SAE features $\mathbf{z} = [z_1, z_2, \ldots, z_D]$ and corresponding correctness scores $\mathbf{y} = [y_1, y_2, \ldots, y_n]$ for $n$ samples, the correlation for each feature $i$ is computed as:

$$r_i = \frac{\mathrm{Cov}(z_i, y)}{\sqrt{\mathrm{Var}(z_i) \cdot \mathrm{Var}(y)}} \tag{1}$$

To handle the computational challenges of large SAE feature dictionaries (typically $10^4$-$10^5$ features), a streaming correlation accumulator is implemented that maintains $O(1)$ memory complexity (see Appendix A.1 for algorithm details). For generation tasks requiring multiple tokens, max-pooling is employed over valid token positions to aggregate feature activations, as empirically validated in our pooling comparison study (Table 3).

### 3.2 COEFFICIENT ESTIMATION FROM POSITIVE OUTCOMES

For each selected feature $i$, we define its steering coefficient as the mean activation over samples with positive task outcomes. Formally:

$$c_i = \frac{1}{|\{j : y_j > 0\}|} \sum_{j : y_j > 0} z_{i,j}. \tag{2}$$

This formulation directly anchors the steering magnitude to the feature's natural activation scale during successful performance. Unlike contrastive-based methods, it leverages the non-negativity of SAE activations (arising from ReLU) (Bricken et al., 2023), thereby avoiding ill-posed subtraction between activation states and ensuring stable, semantically faithful steering. These coefficients are then used at inference time to construct steering vectors that modify the model's residual stream.

### 3.3 INFERENCE-TIME STEERING MECHANISM

At inference time, steering modifies residual stream activations during token generation. For a selected feature $i$ with coefficient $c_i$ and SAE decoder weights $\mathbf{W}_{\mathrm{dec}}$ (its feature direction (Templeton et al., 2024)), the steering vector $\mathbf{v}_{\mathrm{steer}} = c_i \cdot \mathbf{W}_{\mathrm{dec}}[:, i]$ is added to the residual stream, where correlation $r_i$ identifies *which* features to select and coefficient $c_i$ determines *how much* to steer. We apply steering exclusively to generation-time positions, rather than uniformly across all tokens (Soo et al., 2025) or restricted to the final token (Luo et al., 2024; Rimsky et al., 2024). Formally, for a prompt with $n$ tokens:

$$\mathbf{x}'_t = \begin{cases} \mathbf{x}_t & \text{if } t < n \\ \mathbf{x}_t + \sum_{i \in \mathcal{F}} c_i \cdot \mathbf{W}_{\mathrm{dec}}[:, i] & \text{if } t \geq n \end{cases} \tag{3}$$

where $\mathcal{F}$ denotes the set of selected features, $t$ is the token position, and steering begins at the last prompt token ($t = n$) whose residual stream is used to generate the first new token. Since many benchmarks involve multi-token generations, this raises the question of how to aggregate activations across tokens when computing correlations and coefficients, which we address next.

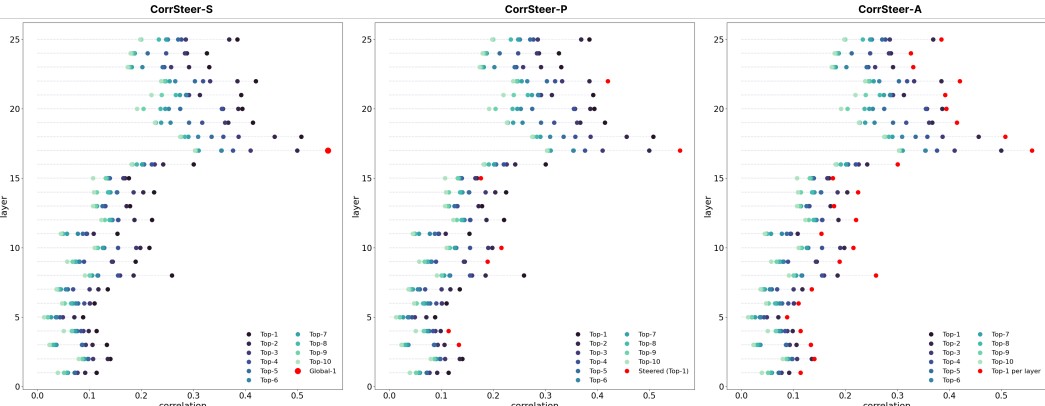

Figure 2: Comparison of features selected by CorrSteer-S, CorrSteer-A, and CorrSteer-P on BBQ (disambiguous) across all Gemma-2 2B layers. Red points denote selected features.

### 3.4 POOLING STRATEGY FOR FEATURE AGGREGATION.

Two pooling strategies are available for coefficient and correlation calculations: max-pooling and mean-pooling. For multi-token generation tasks, max-pooling consistently outperforms mean-pooling, as empirically demonstrated in Table 3, likely due to its better capture of peak feature activations relevant to task success. However, for coefficient calculation in longer generation tasks such as GSM8K reasoning, mean-pooling is preferred as max-pooling produces excessively large coefficient values. Applying these large coefficients to every generated token degrades performance, leading to the adoption of mean-pooling for reasoning tasks. Having established how features are aggregated across tokens, we next turn to how features are selected across layers.

### 3.5 AUTOMATED MULTI-LAYER FEATURE SELECTION

For each layer $\ell$, we extract SAE activations from the residual stream and rank features by their correlation with task performance. We consider both a *global view* aggregating correlations across layers and a *layer-wise view* that preserves layer-specific structure. Based on these perspectives, we implement three fully automated strategies (no hyperparameter tuning required):

- **CorrSteer-S.** Select the single most positively correlated feature across all layers (global view). This minimal variant tests whether a single feature suffices for causal performance improvements.
- **CorrSteer-A.** Select the top positively correlated feature from each layer. This design probes whether layer-wise features collectively form circuits that enhance task performance.
- **CorrSteer-P.** Begin with CorrSteer-A and apply validation-based pruning, retaining only those features that improve over the non-steered model. This enables finer-grained subcircuit analysis.

Only positively correlated features are retained, as ablation experiments confirm that negatively correlated features consistently degrade performance (Table 3). Formal mathematical definitions of these variants are provided in Appendix A.2. Figure 2 illustrates these strategies on the BBQ (disambiguous) task across all layers of Gemma-2 2B, highlighting how CorrSteer-S, CorrSteer-A, and CorrSteer-P differ in terms of selected feature distribution (red points). While CorrSteer-S focuses on a single dominant signal, CorrSteer-A distributes selections across layers, and CorrSteer-P prunes this set to retain only features that yield improvements. These differences highlight distinct trade-offs in global versus layer-wise selection. However, feature selection may also introduce unintended side effects, which we address next.

### 3.6 QUANTIFYING SIDE EFFECTS VIA SER

Correlation-based feature selection risks capturing spurious associations rather than causal drivers, leading to unintended degradations. We quantify this with the *Side Effect Ratio (SER)*, defined as

$$\text{SER} = \frac{\# \text{ negatively changed answers}}{\# \text{ all changed answers}}. \tag{4}$$

Lower SER indicates more reliable steering, isolating features that improve performance without harmful side effects. This measure does not isolate the side effect of each individual feature; rather, it serves as a combined metric reflecting how well selected features are optimized for the task without degrading the model's original abilities. To reduce side effects, the approach focuses on features activated during generation, under the hypothesis that generation-time activations are more likely causally relevant to output. This inference-time focus is empirically validated by our pooling experiments (Table 3). Additionally, in the multi-layer approach, a validation-based filtering mechanism is introduced (**CorrSteer-P**), retaining only features that demonstrate steering effectiveness.

## 4 EXPERIMENTAL SETUP

Experiments are conducted using Gemma-2 2B (Team, 2024a) and LLaMA-3.1 8B (Team, 2024b) models, paired with their corresponding SAE releases from Gemma Scope (Lieberum et al., 2024) and LLaMA Scope (He et al., 2024), respectively. Both SAE families employ JumpReLU activation (Rajamanoharan et al., 2024). Additionally, the Gemma-2 2B-IT model with SAEs is employed, leveraging the fact that SAEs are typically transferable across fine-tuned models (Kissane et al., 2024), with proven low loss reported in the Gemma Scope paper (Lieberum et al., 2024).

**Evaluation Benchmarks** We evaluate CorrSteer on a suite of benchmarks spanning five categories:

- *Knowledge:* MMLU (Hendrycks et al., 2021) and MMLU-Pro (Wang et al., 2024) test broad-domain expertise under zero-shot settings.
- *Reasoning:* GSM8K (Cobbe et al., 2021) probes multi-step mathematical reasoning ability.
- *Bias:* BBQ (Parrish et al., 2022) measures sensitivity to social bias and stereotypes.
- *Factuality:* SimpleQA (Wei et al., 2024) assesses short-form factual consistency.
- *Safety:* HarmBench (Mazeika et al., 2024) and XSTest (Röttger et al., 2024) evaluate resistance to unsafe or sensitive content generation.

For safety benchmarks, both HarmBench (refusal) and XSTest (overrefusal) evaluate steering ability and contextual understanding.

**Side Effect Evaluation.** We measure Side Effect Ratio (SER) to quantify unintended performance degradations (Table 4). CorrSteer's SER is compared against fine-tuning baselines across question-answering datasets. Additionally, we validate our positive-only feature selection by comparing performance when using negatively correlated features (Table 3). We also assess different pooling strategies to verify that inference-time token selection is optimal (Table 3).

**Pooling Strategies for Feature Aggregation.** To verify that our pooling design in Section 3.4 is robust, we conduct an ablation comparing three strategies for aggregating SAE activations across tokens: (i) *mean-pooling*, which averages activations across tokens; (ii) *all-token pooling*, which aggregates contributions from every position; and (iii) *max-pooling*, which selects the strongest activation. We evaluate these alternatives on GSM8K (reasoning), BBQ (bias), and HarmBench/XSTest (safety), covering both single-token and multi-token generation tasks. This setup isolates the effect of pooling and allows us to test whether CorrSteer's empirically motivated default choices are consistently optimal across task types.

**Feature Interpretability and Transferability Analysis.** Performance-improving features are analyzed post-hoc using Neuronpedia descriptions to examine whether correlation-selected features exhibit semantic coherence (Appendix A.11.1). We analyze whether performance-improving features correspond to meaningful behaviors such as refusal, neutrality, or structured reasoning. Safe/unsafe tendency inspection and task-wise breakdowns test whether CorrSteer activates task-relevant semantics rather than spurious signals. Finally, we probe transferability by evaluating features selected on one benchmark (e.g., MMLU) on others (e.g., BBQ, MMLU-Pro) to test whether our method identifies generalizable circuits (Table 2).

## 5 RESULTS AND DISCUSSION

Table 1 and Table 5 present comprehensive results across evaluation benchmarks. CorrSteer demonstrates improvements across question answering, bias mitigation, and safety benchmarks.

Table 1: Performance comparison across CorrSteer variants and other steering methods on Gemma-2 2B. Results are reported as mean ± standard deviation across 5 random seeds (3 for GSM8K). Within each method category, the best results are highlighted in **bold**, and the second-best results are highlighted in *italics*.

| Method | MMLU | MMLU-Pro | SimpleQA | BBQ Ambig | BBQ Disambig | HarmBench | XSTest | GSM8K |
|---|---|---|---|---|---|---|---|---|
| ***CorrSteer Variants*** | | | | | | | | |
| *Non-steered* | 52.21 ± 0.04 | 30.40 ± 0.21 | *3.78 ± 0.17* | 59.46 ± 0.21 | 75.38 ± 0.14 | 46.61 ± 2.78 | 86.35 ± 0.32 | **54.44 ± 0.35** |
| *CorrSteer-S* | 52.99 ± 0.47 | 30.38 ± 0.08 | 3.68 ± 0.07 | 62.39 ± 0.02 | 75.70 ± 0.01 | 46.61 ± 0.76 | *86.77 ± 0.48* | *53.63 ± 0.72* |
| *CorrSteer-P* | *54.70 ± 1.22* | *30.63 ± 0.13* | **3.80 ± 0.14** | **66.00 ± 2.15** | 76.48 ± 0.64 | **66.08 ± 20.20** | 86.46 ± 0.37 | 53.10 ± 0.74 |
| *CorrSteer-A* | **55.48 ± 0.59** | **30.93 ± 0.19** | 3.74 ± 0.07 | 62.06 ± 0.84 | **76.53 ± 0.23** | *73.75 ± 8.84* | **86.98 ± 1.45** | 40.34 ± 24.43 |
| ***Other Methods*** | | | | | | | | |
| *Fine-tuning* | **55.75 ± 0.09** | **35.32 ± 2.70** | – | – | – | – | – | **47.00 ± 0.33** |
| *SPARE (MI)* | 54.97 ± 0.87 | 30.84 ± 0.18 | *3.72 ± 0.04* | 64.81 ± 2.12 | 76.25 ± 0.59 | **65.43 ± 14.34** | 86.82 ± 0.76 | – |
| *DSG (Fisher)* | 52.81 ± 0.59 | 30.33 ± 0.16 | 3.66 ± 0.06 | 61.75 ± 1.39 | 75.61 ± 0.16 | 45.86 ± 1.76 | 86.35 ± 0.59 | – |
| *CAA* | 55.13 ± 1.00 | 28.01 ± 5.79 | 3.71 ± 0.07 | 62.40 ± 1.07 | **76.32 ± 0.40** | 43.14 ± 28.95 | 72.95 ± 17.50 | – |

## 5.1 Comparison with Baselines

Across benchmarks, CorrSteer-A and CorrSteer-P achieve the strongest results, with CorrSteer-P showing particular dominance in LLaMA-3.1 8B. This can be attributed to the less disentangled nature of LLaMA Scope features under superposition, which necessitates more aggressive pruning. Results on both Gemma-2 2B and LLaMA-3.1 8B confirm consistent improvement patterns. CorrSteer-S/A/P represent ablations of feature selection strategies with single global feature, all-layer, and validation-pruned configurations respectively. For comparison with other SAE steering methods under the same multi-layer setting, we report CorrSteer-A. The correlation-based approach outperforms mutual information (MI) and Fisher information-based methods, supporting the faithfulness of SAE's linear representation. This suggests that linear correlation-based feature extraction aligns with the linear latent space of SAEs, where features are designed to be linearly combined. Existing steering approaches rely on contrastive examples restricted to static contexts, while CorrSteer directly leverages generation-time activations, extending SAE-based steering and achieving practical gains across QA, safety, and bias benchmarks.

Head-to-head comparison with CAA (Rimsky et al., 2024; Turner et al., 2025), DSG (Muhamed et al., 2025), or SPARE (Zhao et al., 2025) is not directly applicable since these methods require contrastive datasets rather than generation-time features. However, for comparison purposes, we apply our generation-time feature selection approach to these methods. For fair comparison, we applied the same test-time features and average positive coefficients across methods, with MI and Fisher information-based methods using substituted feature selection while CAA directly uses correct and incorrect answer activation differences. Furthermore, other methods also show improved performance when adapted to use generation-time features, demonstrating the effectiveness of our generation-time feature selection approach independent of the specific steering mechanism.

While fine-tuning achieves higher raw accuracy, CorrSteer offers advantages in side-effect reduction. On MMLU, CorrSteer-A achieves competitive accuracy (55.48% vs. 55.75%) while halving SER (0.21 vs. 0.41) (Table 1, Table 4). Although fine-tuning outperforms CorrSteer variants in raw accuracy on GSM8K and MMLU-Pro, CorrSteer maintains substantially lower SER across tasks. Moreover, CorrSteer can be layered on top of fine-tuned models as complementary enhancement.

## 5.2 Cross-Task Feature Transferability

To evaluate the transferability of selected features across different tasks, we conduct cross-task steering experiments where features selected for one task are applied to different target tasks, as shown in Table 2. This analysis provides insights into the generalizability of task-specific feature sets.

The results reveal several interesting patterns: (1) MMLU and MMLU-Pro features show reasonable cross-transferability, likely due to their shared multiple-choice format and question-answering patterns, (2) BBQ features demonstrate good transferability to MMLU tasks, suggesting that bias mitigation features capture general question-answering capabilities, and (3) features optimized for specific tasks generally outperform transferred features, validating the importance of task-specific feature selection.

Table 2: Cross-task feature transferability results on Gemma-2 2B. Features selected from source tasks (rows) are applied to target tasks (columns). Results show accuracy (%) with non-steered model performance in parentheses. MMLU-Pro results do not use constrained decoding, achieving 17.56% compared to unconstrained non-steered model (14.00%).

| Source → Target | MMLU | MMLU-Pro | BBQ Disambig | BBQ Ambig |
|---|---|---|---|---|
| **MMLU** | **56.32** (52.23) | **19.67** (14.00) | 74.62 (75.42) | **64.01** (59.10) |
| **MMLU-Pro** | 55.73 (52.23) | 17.56 (14.00) | 76.10 (75.42) | 60.97 (59.10) |
| **BBQ Disambig** | 54.74 (52.23) | 16.11 (14.00) | **76.53** (75.42) | 60.85 (59.10) |
| **BBQ Ambig** | 53.85 (52.23) | 11.01 (14.00) | 76.10 (75.42) | 62.08 (59.10) |

**Feature Collaboration and Circuit Effects** CorrSteer-A demonstrates superior performance in 5 out of 8 tasks, indicating that improvements often emerge from feature collaboration within circuits, even when individual feature steering yields limited benefit. Multi-layer approaches such as CorrSteer-A and CorrSteer-P consistently outperform the single-layer CorrSteer-S, aligning with prior findings on circuit-level interventions (Liu et al., 2024; Zhao et al., 2025).

**Safety and Factuality.** On HarmBench, selected features enhance refusal ability, achieving a 27.2% gain, though this primarily reflects increased refusal rather than fine-grained safety. In contrast, XSTest shows limited gains due to the benchmark's over-refusal bias. This outcome is expected given the static nature of CorrSteer, which cannot easily separate benign from harmful requests. Similarly, on SimpleQA, CorrSteer yields only marginal improvement, confirming that the method enhances adherence to task requirements without introducing external factual knowledge. This is desirable, as it suggests CorrSteer modifies behavior rather than injecting content absent from the base model.

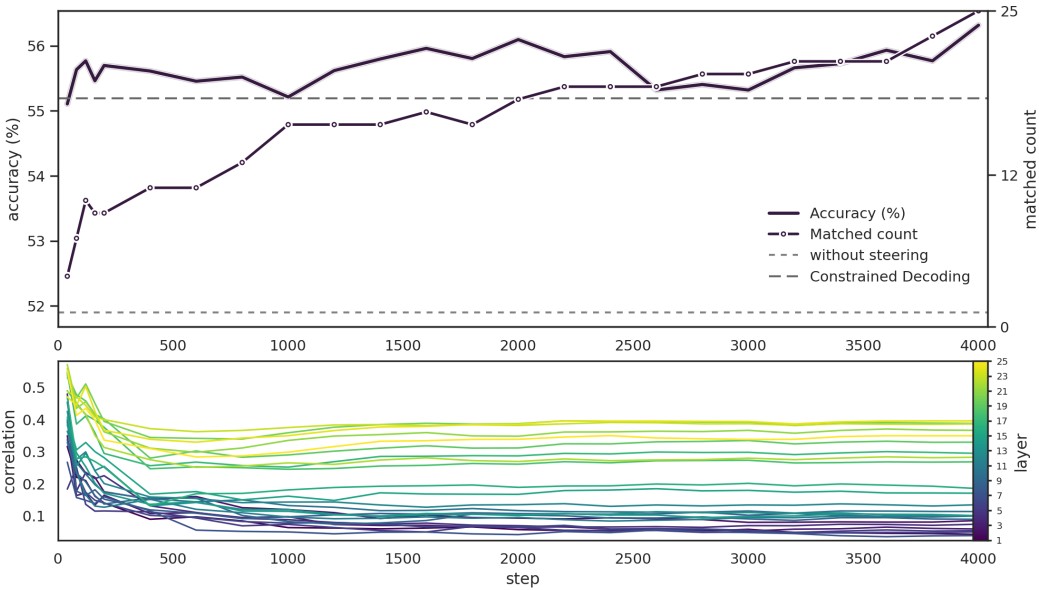

Figure 3: Relation between sample counts and test performance, final matched count of selected features, and most correlated features from each Gemma-2 2B layer. Dotted lines show baseline default LLM performance and constrained decoding performance on MMLU answer options.

## 5.3 EFFICIENCY AND SCALABILITY

CorrSteer serves as an auxiliary mechanism that identifies task-relevant features through generation-time correlations, complementing supervised fine-tuning and remaining effective when applied on top of fine-tuned models. The pipeline is fully automated, requires no hyperparameter tuning, and generalizes across tasks and domains with minimal adjustment. The streaming correlation algorithm

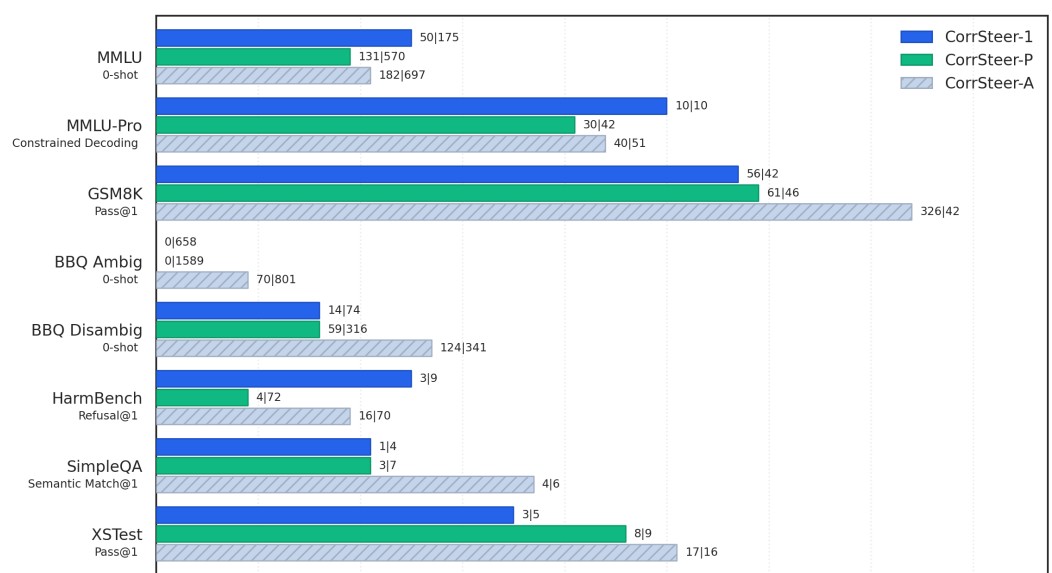

Figure 4: SER comparison between different CorrSteer variants for Gemma-2 2B.

operates with $\mathcal{O}(1)$ memory complexity relative to dataset size, ensuring scalability to large corpora. CorrSteer performs effectively with as few as 100 samples, though stable performance requires approximately 4,000 samples (Appendix A.3). Once steering vectors are extracted, inference requires no SAE dependency, since fixed feature sets and coefficients are sufficient.

**Training Sample Requirements:** As shown in Figure 3, CorrSteer performs effectively even with around 100 training samples, with no substantial improvements beyond 4,000 samples, making it practical for quick deployment. The high variance observed in CorrSteer-A for smaller datasets like GSM8K (1,000 samples) and HarmBench (108 samples) suggests that approximately 4,000 samples are recommended for stable performance.

## 5.4 ABLATION STUDIES

**Pooling Strategies.** As discussed in Section 4, pooling strategy determines how SAE activations are aggregated across tokens. To validate these design choices, we conducted controlled experiments comparing mean-pooling, all-token pooling, and max-pooling across benchmarks covering reasoning (GSM8K), bias (BBQ), and safety (HarmBench, XSTest). The comparison is summarized in Table 3.

Table 3: Ablation studies on pooling strategies and negative correlation features. MMLU-Pro: constrained decoding in (a), unconstrained in (b).

| (a) Pooling strategy comparison | | | | | (b) Positive vs. negative features | | | | |
|---|---|---|---|---|---|---|---|---|---|
| **Task** | **Non** | **Max** | **Mean** | **All** | **Task** | **Non** | **Pos** | **Neg-S** | **Neg-A** |
| MMLU | 52.23 | **56.32** | 56.32 | 52.91 | MMLU | 52.23 | **56.32** | 52.24 | 49.45 |
| MMLU-Pro | 30.30 | **31.00** | 31.00 | 30.16 | MMLU-Pro | 14.00 | **17.56** | 14.24 | 0.66 |
| BBQ Dis. | 75.42 | **76.53** | 76.53 | 75.00 | BBQ Dis. | 75.42 | **76.53** | 75.37 | 12.15 |
| BBQ Amb. | 59.10 | **62.08** | 62.08 | 57.98 | BBQ Amb. | 59.10 | **62.08** | 59.22 | 60.85 |
| HarmBench | 44.64 | **67.50** | 0.00 | 47.14 | HarmBench | 44.64 | **67.50** | 44.64 | 47.86 |
| XSTest | 86.35 | **87.30** | 53.65 | 86.35 | XSTest | 86.35 | **87.30** | 86.35 | 86.67 |
| SimpleQA | 3.63 | **3.80** | 3.76 | 3.73 | SimpleQA | 3.63 | **3.80** | 3.76 | 3.76 |

**Our results reveal clear trends.** On multi-token generation tasks, mean-pooling degrades performance (e.g., HarmBench: 0.00%, XSTest: 53.65%), confirming that averaging dilutes the sparse but informative signals needed for steering. All-token pooling similarly underperforms, suggesting that aggregating contributions from every token introduces substantial noise. By contrast, max-pooling

Table 4: Side Effect Ratio (SER) results on **Gemma-2 2B** across eight benchmarks. Values show mean ± std (5 seeds). Best in **bold**.

| Task | CorrSteer-S | | | CorrSteer-P | | | CorrSteer-A | | |
|---|---|---|---|---|---|---|---|---|---|
| | SER | NEG | POS | SER | NEG | POS | SER | NEG | POS |
| MMLU | $0.25_{\pm0.06}$ | $50_{\pm11}$ | $175_{\pm101}$ | $\mathbf{0.19}_{\pm0.02}$ | $131_{\pm23}$ | $570_{\pm72}$ | $0.21_{\pm0.01}$ | $182_{\pm29}$ | $697_{\pm109}$ |
| MMLU-Pro | $0.50_{\pm0.08}$ | $10_{\pm2}$ | $10_{\pm1}$ | $\mathbf{0.41}_{\pm0.03}$ | $30_{\pm8}$ | $42_{\pm6}$ | $0.44_{\pm0.02}$ | $40_{\pm1}$ | $51_{\pm5}$ |
| GSM8K | $\mathbf{0.57}_{\pm0.01}$ | $56_{\pm32}$ | $42_{\pm22}$ | $0.59_{\pm0.10}$ | $61_{\pm33}$ | $46_{\pm27}$ | $0.74_{\pm0.31}$ | $326_{\pm371}$ | $42_{\pm23}$ |
| BBQ-Ambig | $\mathbf{0.00}_{\pm0.00}$ | $0_{\pm0}$ | $658_{\pm11}$ | $\mathbf{0.00}_{\pm0.00}$ | $0_{\pm0}$ | $1589_{\pm134}$ | $0.09_{\pm0.09}$ | $70_{\pm66}$ | $801_{\pm156}$ |
| BBQ-Disambig | $0.16_{\pm0.02}$ | $14_{\pm1}$ | $74_{\pm5}$ | $\mathbf{0.16}_{\pm0.05}$ | $59_{\pm20}$ | $316_{\pm12}$ | $0.27_{\pm0.05}$ | $124_{\pm21}$ | $341_{\pm41}$ |
| HarmBench | $0.25_{\pm0.13}$ | $3_{\pm2}$ | $9_{\pm5}$ | $\mathbf{0.09}_{\pm0.07}$ | $4_{\pm3}$ | $72_{\pm26}$ | $0.19_{\pm0.27}$ | $16_{\pm24}$ | $70_{\pm30}$ |
| SimpleQA | $\mathbf{0.21}_{\pm0.18}$ | $1_{\pm2}$ | $4_{\pm3}$ | $0.21_{\pm0.03}$ | $3_{\pm3}$ | $7_{\pm4}$ | $0.37_{\pm0.03}$ | $4_{\pm2}$ | $6_{\pm3}$ |
| XSTest | $\mathbf{0.35}_{\pm0.11}$ | $3_{\pm1}$ | $5_{\pm2}$ | $0.46_{\pm0.08}$ | $8_{\pm4}$ | $9_{\pm1}$ | $0.51_{\pm0.10}$ | $17_{\pm4}$ | $16_{\pm7}$ |

| Task | MI (SPARE) | | | Fisher (DSG) | | | CAA | | | Fine-tuning | | |
|---|---|---|---|---|---|---|---|---|---|---|---|---|
| | SER | NEG | POS | SER | NEG | POS | SER | NEG | POS | SER | NEG | POS |
| MMLU | $\mathbf{0.20}$ | 138 | 542 | 0.42 | 55 | 40 | 0.27 | 186 | 515 | 0.41 | 1108 | 1616 |
| MMLU-Pro | $\mathbf{0.43}$ | 38 | 91 | 0.60 | 6 | 4 | 0.55 | 42 | 35 | 0.46 | 357 | 418 |
| GSM8K | 0.63 | 126 | 73 | $\mathbf{0.58}$ | 29 | 50 | 1.00 | 722 | 0 | 0.65 | 213 | 116 |
| BBQ Ambig | $\mathbf{0.00}$ | 5 | 1099 | 0.46 | 39 | 45 | 0.20 | 214 | 1077 | – | – | – |
| BBQ Disambig | $\mathbf{0.17}$ | 16 | 80 | 0.52 | 21 | 44 | 0.62 | 1014 | 612 | – | – | – |
| HarmBench | 0.71 | 53 | 22 | $\mathbf{0.21}$ | 4 | 15 | 1.00 | 132 | 0 | – | – | – |
| SimpleQA | $\mathbf{0.33}$ | 6 | 12 | 0.52 | 12 | 11 | 0.64 | 77 | 43 | – | – | – |
| XSTest | 0.67 | 20 | 10 | $\mathbf{0.32}$ | 13 | 28 | 0.88 | 51 | 7 | – | – | – |

consistently outperforms alternatives across tasks, capturing salient activations while filtering out irrelevant ones. These findings validate our choice of max-pooling as the default aggregation strategy for correlation-based feature selection and steering.

**Negative Correlation Features.**   To validate our design choice of using only positively correlated features, we conduct ablation experiments using negatively correlated features for steering. Table 3 compares single-layer negative steering (Negative-S) and multi-layer negative steering (Negative-A) against CorrSteer-A. Negatively correlated features, applied by subtracting their directions from the residual stream, provide minimal improvement in single-layer steering and severe degradation in multi-layer steering. Notably, MMLU-Pro drops to 0.66% and BBQ Disambig to 12.15% with multi-layer negative steering, confirming that negative correlations often represent spurious patterns rather than causal relationships. In SAE's sparse space, features can activate on negative samples while remaining inactive on positive samples, introducing harmful directions when subtracted. This validates our positive-only approach, which aligns with the non-negative space of SAE activations. Additional ablation studies, including raw activation steering, SAE decoder bias effects, and coefficient-scaling analysis, are provided in Appendix A.6.

## 5.5   SIDE EFFECT TRADE-OFFS

Table 4 and Figure 4 show CorrSteer achieves lower or the same SER compared to other methods while preserving accuracy. CorrSteer-P and CorrSteer-S achieve lower SER than CorrSteer-A, with CorrSteer-P offering the best balance. CorrSteer-S minimizes SER in safety tasks, though single-layer feature quality occasionally limits performance. Side effect patterns also vary with generation length: single-token tasks (MMLU, BBQ) show lower SER, while multi-token generation tasks accumulate more side effects over longer horizons. Positive-only SAE steering methods (CorrSteer, MI, Fisher) exhibit lower SER than fine-tuning, while CAA shows higher SER, which we attribute to its contrastive formulation designed for dense activation spaces (Rimsky et al., 2024).

## 5.6   FEATURE INTERPRETABILITY AND TRANSFERABILITY

Selected features align with task requirements: structured output features dominate multiple-choice benchmarks (MMLU, BBQ), refusal-related features drive safety improvements (HarmBench), and

domain-specific semantics contribute to specialized evaluations. Post-hoc analysis via Neuronpedia descriptions further supports their semantic relevance. Feature activation frequencies vary across tasks, with performance gains tracking activation dynamics (Appendix 7). Mathematical features also emerge across tasks, including bias and safety, consistent with findings that math-oriented pre-training improves broad accuracy (Shao et al., 2024).

For BBQ features in LLaMA-3.1 8B (full list in Appendix A.11.2), positively correlated features emphasize neutrality and balance:

- `L15/25166` **themes of neutrality and balance in discourse** (coeff: 0.259, corr: 0.433)
- `L25/10753` **expressions of perception or belief in social dynamics** (coeff: 1.147, corr: 0.428)

Negatively correlated features on Gemma-2 2B for BBQ capture generic recognition patterns rather than task-specific semantics (full list in Appendix A.11.1):

- `L8/8123` **questions asking for correctness of options** (coeff: 3.725, corr: -0.133)
- `L17/9134` **choice-related phrases and expressions of preference** (coeff: 2.379, corr: -0.451)
- `L19/15745` **decision-making and choice expressions in social contexts** (coeff: 9.740, corr: -0.464)

These results suggest that task-specific semantic features contribute more to accuracy than meta-cognitive recognition features. Our ablation further confirms that SAE-based sparse feature selection outperforms raw activation steering (Table 7).

**Feature Set Transferability.** Cross-task experiments show that MMLU features transfer well, outperforming task-specific features on BBQ Ambig and performing comparably on MMLU-Pro (Table 2). This suggests that certain feature sets capture reasoning patterns shared by multiple-choice benchmarks.

**Task-Level Circuit and Spurious Correlation.** CorrSteer's multi-layer steering relates to circuit discovery research (Olah et al., 2020; Elhage et al., 2021). While prior work isolates task-specific circuits (Conmy et al., 2023; Marks et al., 2025; Ameisen et al., 2025; Lindsey et al., 2025; Sun, 2025), our steering vectors act as additive subgraphs across layers. Restricting feature selection to generation-time activations reduces spurious correlations, and interventions consistently improve performance (Table 1, Table 5), indicating the effectiveness of the selected feature sets.

**Correlation for Selection, Intervention for Causality.** CorrSteer employs correlation as a feature selection mechanism, then establishes causal relationships through direct steering interventions within the controlled LLM computational graph. Unlike spurious correlations with uncontrolled confounding variables, correlations within LLM circuits can be directly validated through residual stream intervention. The consistent performance improvements across tasks (Table 1, Table 5) demonstrate causal influence of selected features.

## 6 CONCLUSION AND LIMITATIONS

This work introduces CorrSteer, a fully automated correlation-driven pipeline that enables generation-time discovery of steering-effective SAE features. By correlating task performance with specific activation patterns during inference, our method extends SAE-based steering to generation-time tasks using correctness signals, eliminating the dependency on contrastive datasets that has limited prior steering approaches. Across eight benchmarks, CorrSteer achieves consistent improvements in question answering, bias mitigation, and safety with minimal computational overhead and reduced side effects, while revealing semantically aligned steering circuits across multiple layers. By leveraging SAE's inherently linear architecture where features are designed to be linearly combined, this design yields interpretable feature combinations without parameter modification, demonstrating linear correlation as an effective approach for mechanistic interpretability.

Despite these advances, limitations remain. The fundamental constraint of steering vectors lies in their static nature, which prevents adaptation to dynamic model behaviors. This particularly affects tasks requiring contextual adaptation or multi-step reasoning, where static steering cannot adequately handle the conditional nature of problem-solving processes. Furthermore, our correlation-based approach exhibits increased performance variance with smaller sample sizes, and the task-optimized features show limited cross-task transferability beyond single-token generation scenarios.

ETHICS STATEMENT

This work investigates correlation-based steering of large language models (LLMs) through Sparse Autoencoder (SAE) features. We have considered the broader impacts of this research in line with the ICLR Code of Ethics.

**Contribute to society and human well-being.** CorrSteer is designed to promote safer and fairer model behavior. We evaluate its impact on reducing harmful generations and mitigating bias, thereby supporting more trustworthy deployment of LLMs.

**Uphold high standards of scientific excellence.** All benchmarks, datasets, and methods will be publicly available upon acceptance, and we will provide full algorithmic details, ablations, and code to enable reproducibility. No human subjects or private data are used.

**Avoid harm.** While CorrSteer improves harmful request refusal, its static nature may lead to over-refusal of benign prompts. We also acknowledge the dual-use potential of steering methods, which could be misapplied to amplify biases or circumvent safety mechanisms.

**Be fair and take action to avoid discrimination.** CorrSteer is explicitly evaluated on social bias benchmarks (e.g., BBQ). Although our method reduces measured biases, residual biases from pretraining data may persist, and further auditing is needed.

**Respect privacy and confidentiality.** This work does not involve personal data, confidential information, or human participants. All resources are used under their respective licenses.

Overall, we believe CorrSteer contributes toward interpretable and responsible control of LLMs, but emphasize the importance of careful auditing, deployment safeguards, and adherence to the ICLR Code of Ethics to minimize risks.

REPRODUCIBILITY STATEMENT

We have taken multiple steps to ensure reproducibility. All benchmarks used in this study (MMLU, MMLU-Pro, BBQ, GSM8K, HarmBench, XSTest, SimpleQA) are publicly available, with dataset splits and sample counts detailed in Section 4. Algorithmic details of correlation computation, coefficient estimation, and inference-time steering are described in Section 3, including pseudocode for the streaming correlation method in Appendix A.1. Hyperparameters for fine-tuning baselines and CorrSteer feature extraction are provided in Section 4. CorrSteer requires no hyperparameter tuning beyond sample size. All code and resources will be publicly released upon acceptance. Results are reported with multiple random seeds (5 seeds, or 3 for GSM8K), and robustness is validated through ablations (Section 5.4) and cross-task transfer experiments (Table 2). These resources provide sufficient information for independent reproduction and extension of our results.

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

## A APPENDIX

### A.1 STREAMING CORRELATION COMPUTATION

To handle the computational challenges of large SAE feature dictionaries (typically $10^4$-$10^5$ features), a streaming correlation accumulator is implemented that maintains $O(1)$ memory complexity:

---

**Algorithm 1** Streaming Correlation Computation

---

Initialize accumulators:

$$\sum x_i = 0, \ \sum x_i^2 = 0, \ \sum x_i y_i = 0, \ \sum y_i = 0, \ \sum y_i^2 = 0, \ n = 0$$

**for** each batch $(\mathbf{X}_{\text{batch}}, \mathbf{y}_{\text{batch}})$ **do**
    Update running sums for each feature dimension
    $n \leftarrow n + |\mathbf{y}_{\text{batch}}|$
**end for**
Compute correlations for each feature $i$:

$$r_i = \frac{n \sum x_i y_i - \sum x_i \sum y_i}{\sqrt{(n \sum x_i^2 - (\sum x_i)^2)(n \sum y_i^2 - (\sum y_i)^2)}}$$

---

This computation maintains $O(1)$ space complexity with respect to sample size, while time complexity is $O(N)$ for $N$ samples, and $O(LD)$ for fixed layer count $L$ and SAE latent dimension $D$.

### A.2 FORMAL DEFINITION OF CORRSTEER VARIANTS

Given $n$ samples with SAE feature activations $z_i^\ell$ at layer $\ell \in \{1, \ldots, L\}$ and feature index $i \in \{1, \ldots, D\}$, and corresponding correctness scores $y \in \mathbb{R}^n$, let $r_i^\ell$ denote the Pearson correlation:

$$r_i^\ell = \frac{\text{Cov}(z_i^\ell, y)}{\sqrt{\text{Var}(z_i^\ell) \cdot \text{Var}(y)}} \tag{5}$$

The three automated feature selection strategies are defined as follows:

**CorrSteer-S (Single):** Selects the globally most correlated feature across all layers:

$$\mathcal{F}_S = \left\{ \arg\max_{(\ell, i)} r_i^\ell : r_i^\ell > 0 \right\} \tag{6}$$

**CorrSteer-A (All layers):** Selects the top correlated feature from each layer:

$$\mathcal{F}_A = \left\{ (\ell, i_\ell^*) : i_\ell^* = \arg\max_i r_i^\ell, \ r_{i_\ell^*}^\ell > 0, \ \forall \ell \in \{1, \ldots, L\} \right\} \tag{7}$$

**CorrSteer-P (Pruned):** Starts with $\mathcal{F}_A$ and applies validation-based pruning:

$$\mathcal{F}_P = \left\{ (\ell, i) \in \mathcal{F}_A : \text{Acc}_{\text{val}}(\mathcal{F}_{\{(\ell, i)\}}) > \text{Acc}_{\text{val}}(\emptyset) \right\} \tag{8}$$

where $\text{Acc}_{\text{val}}(\mathcal{F})$ denotes validation accuracy when steering with feature set $\mathcal{F}$, and $\emptyset$ represents the non-steered baseline.

At inference time, for the selected feature set $\mathcal{F} \in \{\mathcal{F}_S, \mathcal{F}_A, \mathcal{F}_P\}$, the steering at layer $\ell$ and generation position $t \geq n$ is:

$$\mathbf{x'}_t^\ell = \mathbf{x}_t^\ell + \sum_{(\ell', i) \in \mathcal{F}, \ell' = \ell} c_i^\ell \cdot \mathbf{W}_{\text{dec}}^\ell[:, i] \tag{9}$$

where $c_i^\ell = \frac{1}{|\{j : y_j > 0\}|} \sum_{j : y_j > 0} z_{i,j}^\ell$ is the steering coefficient (mean activation over positive outcomes) and $\mathbf{W}_{\text{dec}}^\ell$ is the SAE decoder weight matrix at layer $\ell$.

### A.3   Implementation Details

**Feature Extraction:**   Feature selection employs 4,000 samples across all datasets. For fair comparison, the same samples are used for training fine-tuning models. When datasets contain fewer than 4,000 samples, we use all available data. For datasets without predefined train/validation/test splits, we allocate 27% for training, 3% for validation, and 70% for testing. GSM8K uses 1,000 samples for feature selection with 50 samples reserved for validation.

**Feature Steering:**   Steering interventions are applied at the pre-execution stage of each transformer layer. The first layer is excluded from steering as the token embedding layer predominantly contains spurious correlations unrelated to the target tasks.

**Evaluation Metrics:** For multiple-choice tasks (MMLU, MMLU-Pro, BBQ), exact match accuracy is used under zero-shot evaluation. All results are reported as mean ± standard deviation across multiple random seeds for statistical robustness: 5 seeds for most tasks, 3 seeds for GSM8K. For Gemma-2 2B, the non-steered MMLU performance (52.23%) is lower than the Gemma-2 2B-IT 5-shot result (56.1%) reported in the original Gemma paper due to the zero-shot setting and lack of in-context learning examples. For safety benchmarks, 1 - ASR (Attack Success Rate) is computed using a small refusal-detection language model. SimpleQA performance is measured using a small STS language model to match the expected answer, with more details in Appendix A.4.

A standard train-validation-test split is used for the CorrSteer pipeline. The training dataset is used to extract correlated SAE features, and the validation dataset is used to filter the most correlated features. The test dataset is used to evaluate the performance of the CorrSteer pipeline. Detailed configurations are provided in Appendix A.3.

**Fine-tuning**   Fine-tuning hyperparameters are determined through empirical experimentation across tasks and dataset sizes. Fine-tuning is performed using AdamW optimizer with learning rate 1e-5 (reduced to 5e-6 for small datasets <2000 samples), weight decay 0.01, and gradient clipping at norm 1.0. The training schedule includes 3% warmup steps followed by cosine annealing decay. Training proceeds for one epoch with 4,000 samples, using exact target supervision where prompt tokens are masked with -100 labels and only target spans contribute to the loss.

### A.4   Generation Benchmark Results

**Evaluation Models:** Two specialized models are employed for evaluation. The DistillRoBERTa model[1] is used to identify the rejection of harmful requests, while the ModernBERT STS model[2] is used for matching generated answers against expected responses.

### A.5   Additional Results

Table 5: Performance comparison between non-steered model and CorrSteer variants across BBQ, MMLU, MMLU-Pro, HarmBench, SimpleQA, and XSTest on LLaMA-3.1 8B. Results show accuracy (%) under zero-shot evaluation (single-shot for BBQ).

| Task | Non-steered | Corrsteer-S | CorrSteer-P | CorrSteer-A |
|------|-------------|-------------|-------------|-------------|
| BBQ Ambig | 83.97 | 83.98 | **87.10** | 86.83 |
| BBQ Disambig | 90.07 | 90.13 | **90.33** | 90.30 |
| HarmBench | 0.71 | 0.36 | 15.71 | **17.86** |
| MMLU | 61.41 | 61.51 | **61.73** | 61.71 |
| MMLU-Pro | 32.13 | 32.55 | **35.08** | 34.71 |
| SimpleQA | 0.43 | **0.51** | 0.43 | 0.43 |
| XSTest | 61.27 | **62.22** | **62.22** | 58.41 |

---

[1] https://huggingface.co/protectai/distilroberta-base-rejection-v1
[2] https://huggingface.co/dleemiller/ModernCE-base-sts

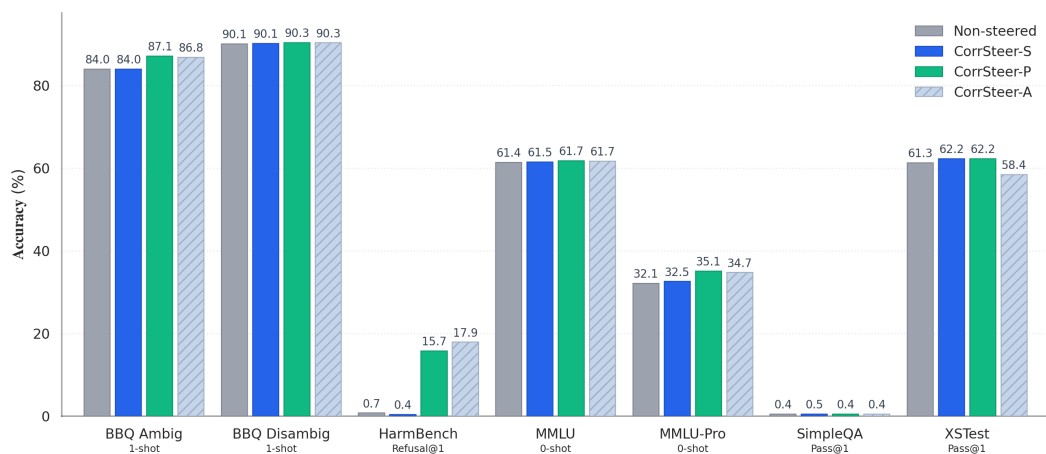

Figure 5: Benchmark performance of CorrSteer variants compared with the non-steered model on LLaMA-3.1 8B.

**Task-Specific Analysis**   *MMLU:* The global method selects features related to structured output formatting, addressing Gemma-2 2B's tendency to generate tokens outside the required A/B/C/D options. Post-steering, this hallucination issue is largely resolved.

*MMLU-Pro:* A similar issue occurs more severely due to the 10 options in MMLU-Pro. Constrained decoding, which samples tokens exclusively from available options, is applied to improve the model's authentic capability, resulting in performance that remains higher than the non-steered model, with CorrSteer-A achieving maximum performance.

*BBQ:* Similar improvements in format adherence are observed, with selected features promoting appropriate response structure.

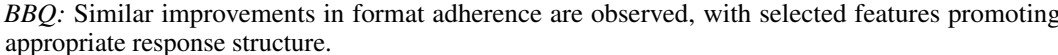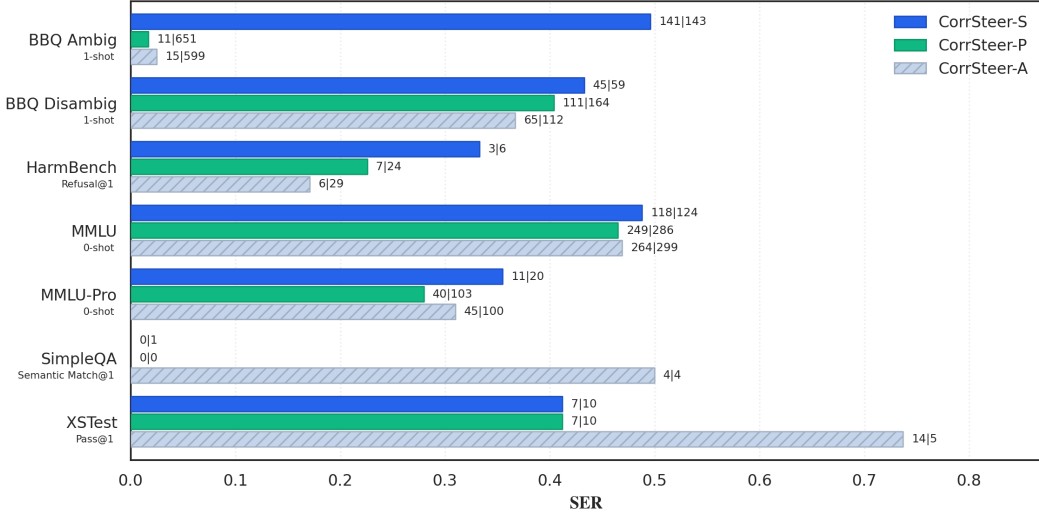

Figure 6: SER comparison across datasets between different CorrSteer variants on LLaMA-3.1 8B.

**Feature Frequency Analysis**   We observe a strong correlation between feature activation frequency and CorrSteer's performance improvements across tasks. As demonstrated in Figure 7, HarmBench exhibits consistently high activation frequencies across all layers, while SimpleQA shows frequencies approaching zero.

This pattern contrasts with the typical sparse activation nature of SAE features, where low frequency activation (below 5%) is considered normal and interpretable, while higher frequencies typically indicate non-interpretable (Stolfo et al., 2025; Smith et al., 2025). However, discovering task-specific features with near-100% activation frequency suggests these features are deeply related to the task

Table 6: Side Effect Ratio (SER) analysis for CorrSteer variants on LLaMA-3.1 8B across different benchmarks. SER values closer to 0 indicate better safety performance.

| Task | Corrsteer-S | | | CorrSteer-P | | | CorrSteer-A | | |
|---|---|---|---|---|---|---|---|---|---|
| | SER | neg | pos | SER | neg | pos | SER | neg | pos |
| BBQ Ambig | 0.496 | 141 | 143 | **0.017** | 11 | 651 | 0.025 | 15 | 599 |
| BBQ Disambig | 0.433 | 45 | 59 | 0.404 | 111 | 164 | **0.367** | 65 | 112 |
| HarmBench | 0.333 | 3 | 6 | 0.226 | 7 | 24 | **0.171** | 6 | 29 |
| MMLU | 0.488 | 118 | 124 | **0.465** | 249 | 286 | 0.469 | 264 | 299 |
| MMLU-Pro | 0.355 | 11 | 20 | **0.280** | 40 | 103 | 0.310 | 45 | 100 |
| SimpleQA | **0.000** | 0 | 1 | - | 0 | 0 | 0.500 | 4 | 4 |
| XSTest | 0.412 | 7 | 10 | 0.412 | 7 | 10 | 0.737 | 14 | 5 |

requirements, resulting in substantial performance improvements for such tasks. Even for tasks with lower feature frequencies, CorrSteer maintains its advantage by preserving low SER values.

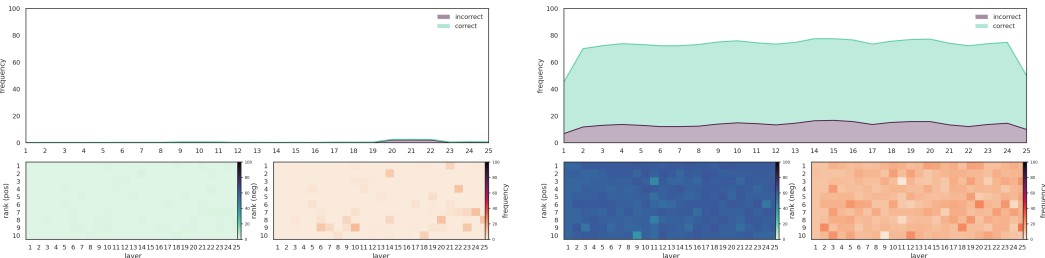

Figure 7: Frequency of activation samples across layers of Gemma-2 2B for SimpleQA (left) and HarmBench (right) tasks.

## A.6 ADDITIONAL ABLATION STUDIES

Table 7: Performance comparison between raw activation steering and SAE-decoded steering on Gemma-2 2B. Decoding adds SAE decoder bias term for the first layer, while Decoding-A adds multi-layer feature directions as CorrSteer-A.

| Task | Non-steered | Raw Activation | Decoding-S | Decoding-A | CorrSteer-A |
|---|---|---|---|---|---|
| MMLU | 52.23 | 49.85 | 55.38 | 54.38 | **56.32** |
| MMLU-Pro | 30.30 | 27.17 | 29.79 | 29.93 | **31.00** |
| BBQ Disambig | 75.42 | 75.71 | **77.00** | 75.03 | 76.53 |
| BBQ Ambig | 59.10 | 58.42 | 54.00 | 55.76 | **62.08** |

**Raw Activation Steering** To validate the effectiveness of SAE-based sparse feature selection, we compare steering performance using raw residual stream activations. The results demonstrate a clear performance hierarchy: CorrSteer-A > SAE Decoding > Raw Activation across all evaluated tasks, which is explainable by Superposition Hypothesis (Elhage et al., 2022). One exception occurred in BBQ Disambig, where Decoding-S shows better performance than CorrSteer-A. However, Decoding-S failed to show robustness across benchmarks, frequently degrading performance while CorrSteer-A shows consistent performance across all tasks.

**SAE Decoder Bias** Adding SAE decoder bias terms alongside selected features improves performance only at single-token generation tasks (BBQ, MMLU, MMLU-Pro). This effect appears related to attention sink mechanisms (Xiao et al., 2024), where increased residual stream norms amplify attention patterns in subsequent layers, acting similar to "response prefix" (Hazra et al., 2025). For constrained generation tasks, this norm amplification reduces hallucination by strengthening adherence to output format constraints. However, this enhancement is incompatible with multi-layer

steering and diminishes when applied across multiple layers or tokens, with excessive application potentially causing model collapse.

## A.7 TEXT CLASSIFICATION VALIDATION

To validate the effectiveness of correlation-based feature selection, we conduct controlled experiments on text classification tasks where ground truth labels provide clear supervision signals. The experiments utilize GPT-2 (Radford et al., 2019) with publicly available SAEs from Bloom et al. (Bloom, 2024) on the bias-focused text classification dataset EMGSD (King et al., 2024).

For each bias category, we extract the most correlated features using max-pooling over all text tokens, then apply steering by either adding positively correlated features or subtracting negatively correlated features. Steering effectiveness is evaluated using the same classifier employed in the original dataset.

Table 8: Bias steering effectiveness across different demographic categories on EMGSD dataset. Mitigation reduces bias scores, while amplification increases them.

| Category | Mitigation (Fairness ↑) | | Amplification (Bias ↑) | |
|---|---|---|---|---|
| | Non-steered | CorrSteer | Biased | CorrSteer |
| Gender | 0.177 | 0.616 | 0.897 | 0.922 |
| LGBTQ+ | 0.091 | 0.561 | 0.941 | 0.882 |
| Nationality | 0.125 | 0.732 | 0.937 | 0.945 |
| Profession | 0.128 | 0.625 | 0.890 | 0.921 |
| Race | 0.308 | 0.769 | 0.846 | 0.846 |
| Religion | 0.109 | 0.655 | 0.945 | 0.928 |

Results demonstrate that correlation-selected features provide effective steering control across all demographic categories (Table 8). For mitigation, CorrSteer surpasses the non-steered model across categories by improving fairness scores. For amplification, CorrSteer generally increases bias relative to the biased non-steered model, with the LGBTQ+ row as an exception to be audited.

## A.8 FRAMEWORK IMPLICATIONS

CorrSteer leverages generation-time activations for multi-token, multi-layer SAE-based steering, and our experiments are enabled by Gemma Scope (Lieberum et al., 2024) and LLaMA Scope (He et al., 2024), the only open releases providing SAEs across all residual stream layers.

The proposed framework demonstrates the practical utility of SAE in real-world LLM inference, addressing critical concerns such as safe reasoning, bias mitigation, and resistance to jailbreaking. This research demonstrates that SAE-based control mechanisms offer a promising direction for both understanding and improving LLM behavior. The framework's ability to operate through an interpretable interface while maintaining or improving model performance suggests a concrete path toward safer, more transparent AI.

## A.9 COEFFICIENT AND CORRELATION SCALE DIFFERENCES BETWEEN MODELS

The observed differences in coefficient and correlation scales between Gemma-2 2B and LLaMA-3.1 8B stem from two primary factors:

**SAE Architecture Differences:** LLaMA-Scope employs TopK SAEs (Gao et al., 2024), which enforce fixed sparsity through top-k selection, while Gemma-Scope uses JumpReLU SAEs with adaptive thresholding.

**Model and SAE Capacity Differences:** The models differ in base model size (2B vs 8B parameters) and SAE dictionary capacity (16K vs 32K features).

## A.10 Layer-wise Correlation Patterns

Analysis of per-layer correlation values reveals that task-specific features emerge progressively across network depth. For example, Gemma-2 2B on MMLU exhibits correlation increases from 0.140 (Layer 1) to 0.336 (Layer 25), while LLaMA-3.1 8B on BBQ Disambig shows growth from 0.086 (Layer 1) to 0.297 (Layer 20). This hierarchical emergence suggests that later transformer layers encode more task-relevant representations. The trend is attenuated in tasks with lower overall steering effectiveness (SimpleQA, XSTest), where feature-outcome correlations remain weak across all layers. Complete layer-wise correlation values and feature lists are provided below.

## A.11 Complete Feature Lists

This section presents the complete feature lists for each task, showing the top-1 features aggregated from all layers. Each feature is labeled with the format L{layer}/{index} to identify its layer and index position. Features selected by CorrSteer-P after pruning are highlighted in **bold**.

Each feature entry includes the feature description along with its coefficient and correlation value. SAE feature descriptions are obtained through the Neuronpedia API (https://www.neuronpedia.org/), providing automated semantic interpretations of selected features. Feature indices are hyperlinked to their corresponding Neuronpedia pages for detailed analysis.

Feature descriptions that are well-aligned with the target task are highlighted in **bold**, and the highest correlations for each task are also emphasized in **bold**. Following each layer's highest correlated feature, we include additional relevant features listed below. As discussed in Appendix A.10, examining these correlation values across layers reveals that task-specific features generally emerge more strongly in later layers.

### A.11.1 Gemma-2B

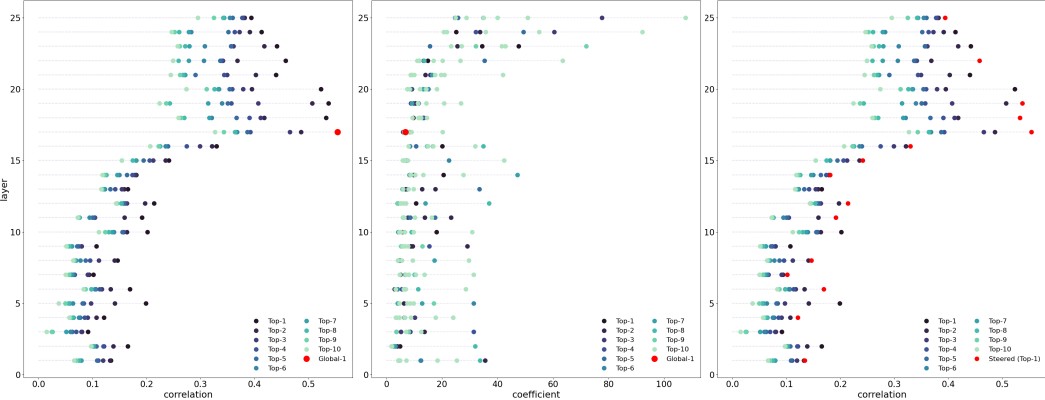

Figure 8: Top correlated features with selected features from CorrSteer-P with BBQ ambig on coefficient in each layer of Gemma-2 2B.

**BBQ (Ambiguous)**

- L1/6088 specific formatting or structural elements within text, such as timestamps and code (coeff: 2.280, corr: 0.134)
- L2/15089 key actions and processes related to achievements and collaboration (coeff: 4.898, corr: 0.166)
- **L3/6151** references to statistical or numerical data in research contexts (coeff: 3.537, corr: 0.091)
- **L4/11047** certain types of mathematical or programming syntax (coeff: 2.854, corr: 0.121)
- L5/7502 expressions of honesty and self-awareness in discourse (coeff: 3.117, corr: 0.199)
- L6/324 structured sentences that present facts, warnings, or errors, often with an emphasis on important details (coeff: 2.886, corr: 0.169)

- L7/4487 the presence of detailed structured elements within a document, such as headings or separators in a legal or formal layout (coeff: 4.996, corr: 0.102)
- L8/4669 special tokens or specific formatting in the text (coeff: 4.378, corr: 0.147)
- **L9/1435** elements related to copyright and licensing information (coeff: 8.737, corr: 0.107)
- **L10/4557** interactions involving guessing or determining the correctness of information (coeff: 4.246, corr: 0.202)
- L11/6144 return statements in code (coeff: 4.347, corr: 0.192)
- L12/15862 punctuation marks and formatting elements in the text (coeff: 2.718, corr: 0.214)
- L13/4379 punctuation symbols and their frequency (coeff: 6.779, corr: 0.165)
- L14/12922 dialogue or conversational exchanges involving questioning and responses (coeff: 1.754, corr: 0.181)
- **L15/12813** medical terms related to respiratory health and conditions (coeff: 3.537, corr: 0.242)
- L16/9006 declarations regarding conflicts of interest and funding in research publications (coeff: 2.606, corr: 0.330)
- **L17/11021** phrases related to scientific research and findings (coeff: 6.777, corr: **0.554**)
- L18/14447 references to medical data and statistics (coeff: 9.667, corr: 0.533)
- L19/11289 assignment and return statements in programming contexts (coeff: 10.429, corr: 0.538)
- L20/2040 occurrences of logical values and conditions in programming or data handling contexts (coeff: 9.166, corr: 0.523)
- L21/8433 keywords related to programming functions and their definitions (coeff: 5.983, corr: 0.440)
- **L22/10377** code snippets that include assignments and return statements (coeff: 14.919, corr: 0.458)
- L23/6394 structured data or code-like formats (coeff: 34.482, corr: 0.442)
- L24/14051 references to education systems and their impact on health initiatives (coeff: 25.098, corr: 0.413)
- L25/12534 references to emotional states or descriptions of personal experiences (coeff: 18.414, corr: 0.394)

*Additional relevant features:*

- L8/8123 questions that ask for truthfulness or correctness regarding options or statements (coeff: 3.725, corr: -0.133)
- L17/9134 choice-related phrases and expressions of preference (coeff: 2.379, corr: -0.451)
- L19/15745 phrases related to decision-making and choice, particularly in the context of parenting and social interactions (coeff: 9.740, corr: -0.464)

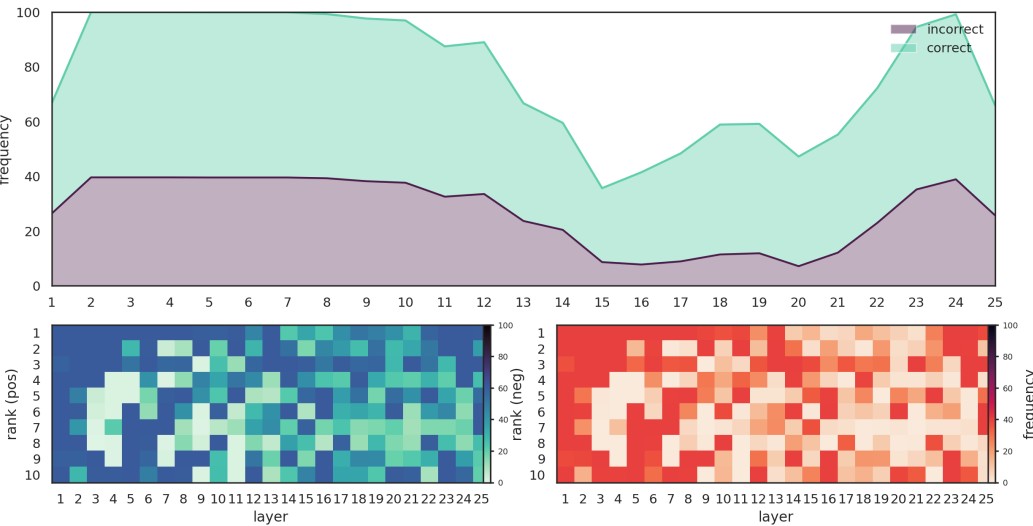

Figure 9: Top correlated features with BBQ ambig on frequency in each layer of Gemma-2 2B.

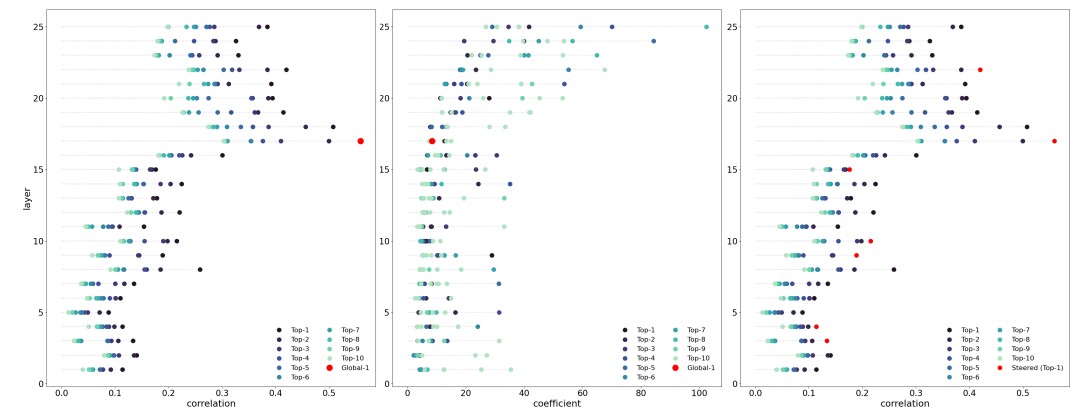

Figure 10: Top correlated features with selected features from CorrSteer-P with BBQ disambig on coefficient in each layer of Gemma-2 2B.

**BBQ (Disambiguous)**

- L1/7001 code structure and elements in programming, particularly related to class and variable definitions (coeff: 2.126, corr: 0.114)
- L2/8432 HTML and JavaScript code related to the Bootstrap framework (coeff: 2.418, corr: 0.140)
- **L3/10179** terms related to health and medical supplements (coeff: 2.383, corr: 0.134)
- **L4/3444** various types of headers, specifically those that denote responses and results within the context of exchanges or interactions (coeff: 2.192, corr: 0.114)
- L5/697 terms related to price dynamics and economic relationships (coeff: 3.766, corr: 0.088)
- L6/2491 references to sources or citations in a document (coeff: 2.618, corr: 0.110)
- L7/6269 references to visual elements such as figures and tables (coeff: 1.293, corr: 0.135)
- L8/5927 mathematical examples and notations (coeff: 3.347, corr: 0.259)
- **L9/7854** structures related to the declaration and manipulation of result variables in a programming context (coeff: 10.475, corr: 0.189)
- **L10/15705** references to file operations and data management in code (coeff: 6.145, corr: 0.215)
- L11/13926 mathematical expressions and calculations (coeff: 8.203, corr: 0.154)
- L12/1085 references to court cases and legal statutes (coeff: 1.839, corr: 0.220)
- L13/536 technical details related to manufacturing processes (coeff: 4.417, corr: 0.178)
- L14/10612 structured data or code snippets related to databases (coeff: 5.030, corr: 0.225)
- **L15/2822** structured data formats or code snippets related to programming (coeff: 1.632, corr: 0.176)
- L16/6602 the presence of specific numerical or coding patterns in data (coeff: 6.773, corr: 0.300)
- **L17/5137** mathematical symbols and functions related to field theories (coeff: 8.483, corr: **0.559**)
- L18/3178 code or programming-related elements (coeff: 7.851, corr: 0.507)
- L19/11641 technical components or elements in code (coeff: 16.336, corr: 0.414)
- L20/12748 **structured data representations and their attributes** (coeff: 28.025, corr: 0.394)
- L21/14337 code-related keywords and method definitions in programming contexts (coeff: 20.453, corr: 0.392)
- **L22/13921** elements related to database structure and definitions (coeff: 18.510, corr: 0.420)
- L23/12349 technical terms related to software or code management (coeff: 5.893, corr: 0.331)
- L24/16355 definitions and mathematical notation in text (coeff: 39.910, corr: 0.326)
- L25/4307 occurrences of programming syntax related to object-oriented structures (coeff: 19.460, corr: 0.384)

*Additional relevant features:*

- L18/1127 references to gender and associated options/choices in forms (coeff: 4.813, corr: 0.207)
- L19/15745 phrases related to decision-making and choice, particularly in the context of parenting and social interactions (coeff: 11.875, corr: 0.226)
- L23/12048 terms related to racism and social injustice (coeff: 2.661, corr: 0.147)

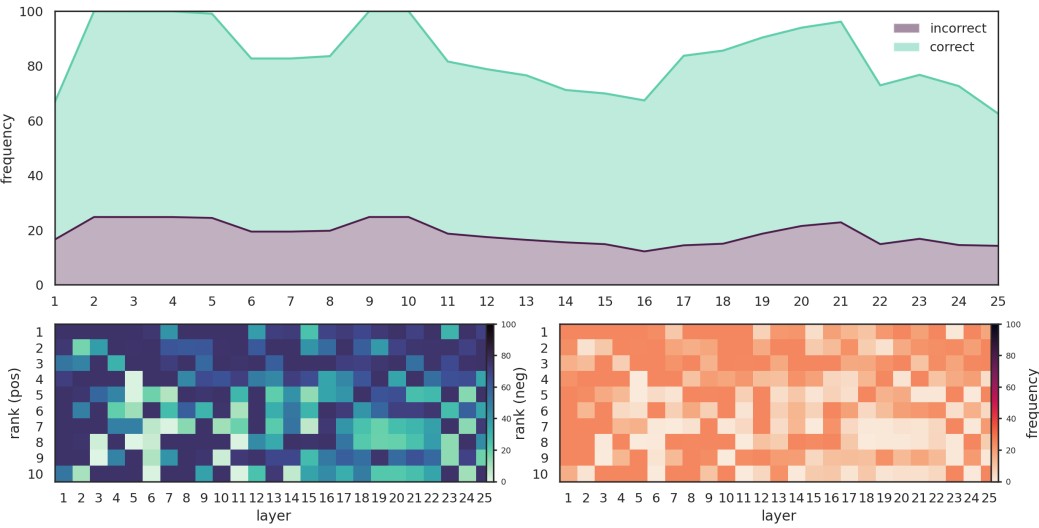

Figure 11: Top correlated features with BBQ disambig on frequency in each layer of Gemma-2 2B.

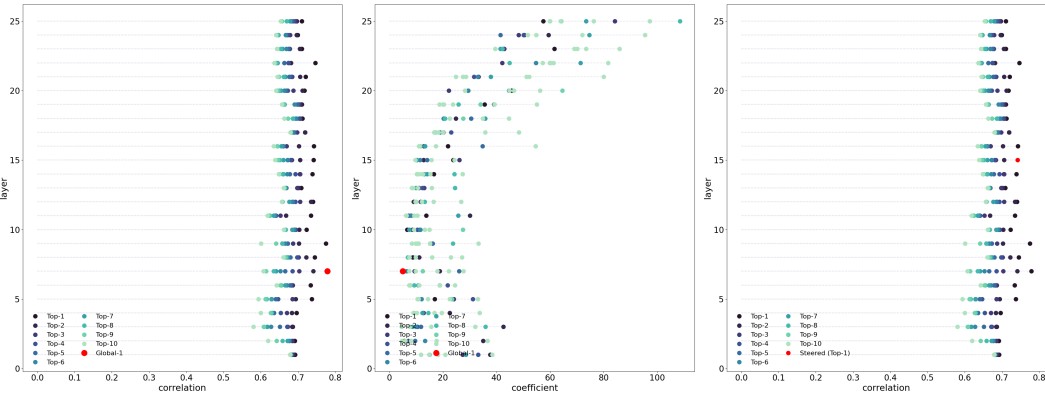

Figure 12: Top correlated features with selected features from CorrSteer-P with HarmBench on coefficient in each layer of Gemma-2 2B.

**HarmBench**

- L1/9572 occurrences of the semicolon character (coeff: 5.206, corr: 0.692)
- L2/6712 references to worship and its related symbols or icons (coeff: 5.699, corr: 0.692)
- L3/16207 syntax elements and formatting in code or mathematical expressions (coeff: 2.583, corr: 0.686)
- L4/3109 forms of the verb "to be" and its variations (coeff: 5.891, corr: 0.696)
- L5/11099 sentences that include personal affirmations or declarations of identity (coeff: 16.934, corr: 0.737)
- L6/12241 instances of the verb "to be" in various forms and their contexts (coeff: 7.338, corr: 0.735)

- L7/11722 **phrases related to legal terms and the rejection of arguments in court cases** (coeff: 5.035, corr: **0.779**)
- L8/8642 expressions of self-identity and subjective experience (coeff: 8.729, corr: 0.745)
- L9/9298 **strongly negative or dismissive opinions about claims and arguments** (coeff: 7.525, corr: 0.775)
- L10/3037 references to legal issues and compliance (coeff: 6.667, corr: 0.723)
- L11/6905 statements of identity and self-description (coeff: 13.810, corr: 0.735)
- L12/12039 phrases related to providing assistance and support (coeff: 5.253, corr: 0.741)
- L13/6715 text that discusses accountability and the need for forgiveness (coeff: 6.992, corr: 0.709)
- L14/2949 statements and phrases related to political criticism and condemnation (coeff: 16.620, corr: 0.739)
- L15/1570 judgments regarding moral and ethical standards related to exploitation and human rights issues (coeff: 23.824, corr: 0.742)
- L16/5113 expressions of personal identity and emotional states (coeff: 21.832, corr: 0.743)
- L17/5887 references to tools and functional capabilities related to programming or software development (coeff: 11.389, corr: 0.720)
- L18/1411 negative statements or denials (coeff: 20.537, corr: 0.712)
- L19/324 phrases related to legal procedures and considerations (coeff: 35.610, corr: 0.710)
- L20/5192 questions that seek clarification or challenge assumptions (coeff: 45.662, corr: 0.718)
- L21/7129 negative sentiments and expressions of doubt or denial (coeff: 33.225, corr: 0.721)
- L22/3311 references to food and culinary experiences (coeff: 19.000, corr: 0.746)
- L23/11246 instances of strong negative sentiment or rejection (coeff: 61.642, corr: 0.711)
- L24/12773 first-person pronouns and references to personal experiences or actions (coeff: 50.332, corr: 0.699)
- L25/3912 **negative sentiments or refusals** (coeff: 57.431, corr: 0.711)

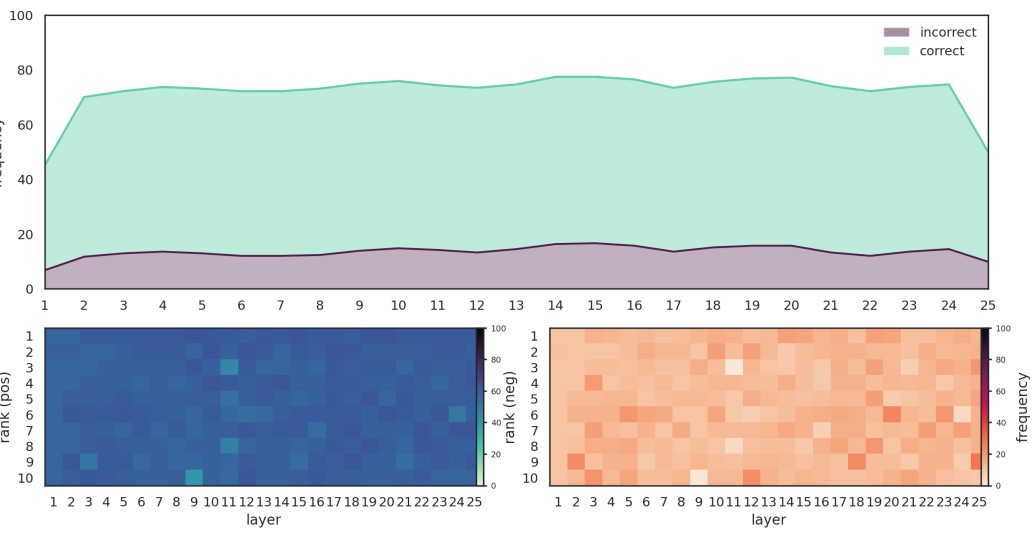

Figure 13: Top correlated features with HarmBench on frequency in each layer of Gemma-2 2B.

**MMLU**

- L1/13714 colons and semicolons used in lists or programming syntax (coeff: 0.403, corr: 0.140)
- L2/6273 specific medical terminology and its implications (coeff: 1.548, corr: 0.175)
- L3/12378 programming-related elements and commands (coeff: 1.094, corr: 0.164)
- L4/11047 certain types of mathematical or programming syntax (coeff: 2.944, corr: 0.225)
- L5/8581 phrases that indicate research findings or results (coeff: 0.077, corr: 0.115)

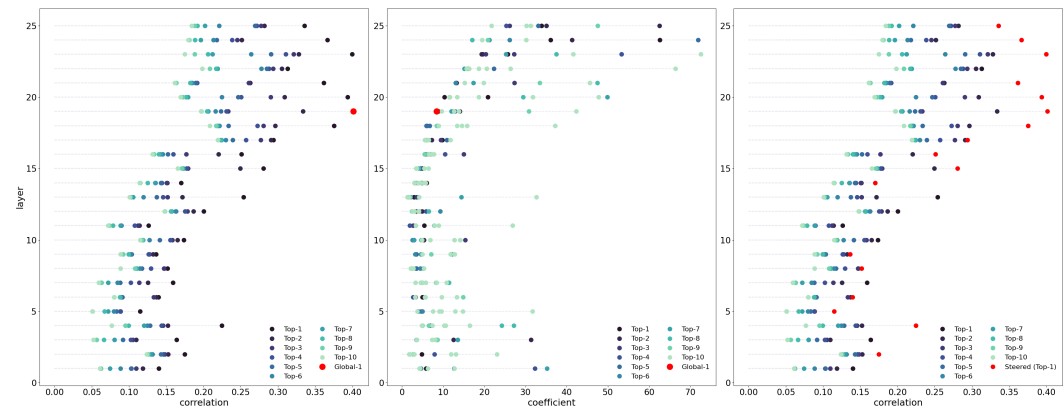

Figure 14: Top correlated features with selected features from CorrSteer-P with MMLU on coefficient in each layer of Gemma-2 2B.

- L6/5275 sentences expressing doubt or conditionality in arguments (coeff: 4.939, corr: 0.140)
- L7/14726 periods and other punctuation marks that signify sentence endings or significant separations in text (coeff: 2.532, corr: 0.159)
- L8/15039 terms related to research methodologies and experimental design (coeff: 0.309, corr: 0.152)
- L9/15654 variations of the word "correct" in various contexts (coeff: 0.414, corr: 0.136)
- L10/11729 coding attributes and properties related to light types in a 3D programming context (coeff: 2.919, corr: 0.174)
- L11/13204 code syntax and structure, particularly related to variable assignments and function calls (coeff: 5.369, corr: 0.126)
- L12/6392 XML-like structured data elements (coeff: 1.033, corr: 0.200)
- L13/12281 mathematical expressions and concepts related to positive values (coeff: 0.919, corr: 0.254)
- L14/7 significant scientific findings and their specific details (coeff: 6.002, corr: 0.170)
- L15/8678 phrases related to announcements or updates (coeff: 4.906, corr: 0.281)
- L16/12421 programming constructs and their structures within code snippets (coeff: 5.593, corr: 0.251)
- L17/13214 error messages and diagnostic codes (coeff: 9.790, corr: 0.294)
- L18/1127 references to gender and associated options/choices in forms (coeff: 4.805, corr: 0.376)
- L19/2174 input fields and value assignments in a form-like structure (coeff: 8.405, corr: **0.402**)
- L20/12748 **structured data representations and their attributes** (coeff: 20.884, corr: 0.394)
- L21/14337 code-related keywords and method definitions in programming contexts (coeff: 13.228, corr: 0.362)
- L22/5939 technical jargon and terminology related to chemistry and biochemistry (coeff: 5.582, corr: 0.313)
- L23/10424 statistical terms and symbols related to data analysis and significance testing (coeff: 25.724, corr: 0.400)
- L24/16355 definitions and mathematical notation in text (coeff: 36.077, corr: 0.367)
- L25/10388 phrases related to health-related actions and topics (coeff: 33.899, corr: 0.336)

**MMLU-Pro**

- L1/9317 phrases related to changes in social and organizational dynamics (coeff: 1.859, corr: 0.169)
- L2/3714 mathematical notation, specifically related to set notation and expressions involving functions (coeff: 0.761, corr: 0.226)

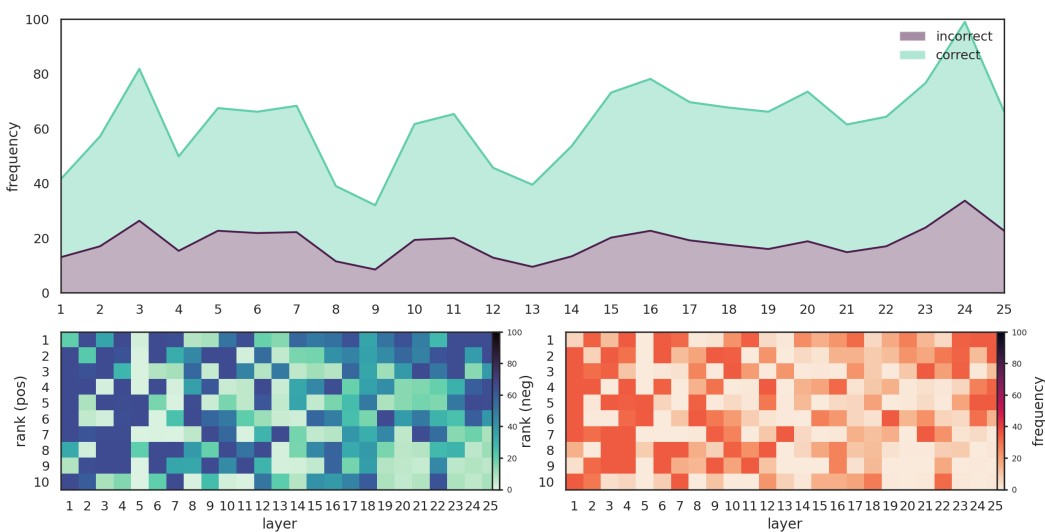

Figure 15: Top correlated features with MMLU on frequency in each layer of Gemma-2 2B.

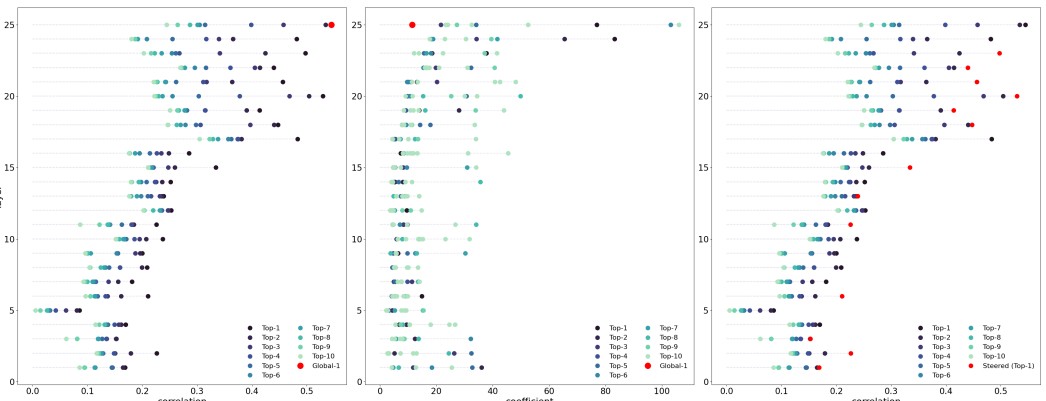

Figure 16: Top correlated features with selected features from CorrSteer-P with MMLU-Pro on coefficient in each layer of Gemma-2 2B.

- L3/11980 statements providing answers or conclusions regarding questions or hypotheses (coeff: 3.699, corr: 0.153)
- L4/15960 terms related to medical procedures and conditions (coeff: 6.817, corr: 0.170)
- L5/7502 expressions of honesty and self-awareness in discourse (coeff: 2.187, corr: 0.086)
- L6/6201 numeric representations of system specifications or configurations (coeff: 14.877, corr: 0.210)
- L7/8790 structured data formats and their attributes (coeff: 1.209, corr: 0.182)
- L8/11297 structured data and programming constructs (coeff: 2.176, corr: 0.209)
- L9/15336 references to mathematical or computational problems and their solutions (coeff: 6.407, corr: 0.200)
- L10/10805 terms related to medical conditions and biological factors (coeff: 1.277, corr: 0.237)
- L11/1909 affirmative or negative responses in the context of questions (coeff: 2.296, corr: 0.226)
- L12/14752 legal and governmental terms related to authority and judgment (coeff: 1.369, corr: 0.253)
- L13/12991 mathematical operations and expressions (coeff: 2.560, corr: 0.239)
- L14/10780 comments and documentation markers in code (coeff: 1.455, corr: 0.252)

- `L15/2262` references to variable declarations and data structures in programming contexts (coeff: 1.183, corr: 0.334)
- `L16/3142` mathematical symbols and notation used in equations (coeff: 5.691, corr: 0.285)
- `L17/1175` mathematical expressions and applications related to programming or data structures (coeff: 3.091, corr: 0.483)
- `L18/682` function declarations and their return types in a programming context (coeff: 3.406, corr: 0.448)
- `L19/11641` technical components or elements in code (coeff: 2.144, corr: 0.414)
- `L20/12748` **structured data representations and their attributes** (coeff: 7.134, corr: 0.529)
- `L21/1944` code structures and syntax related to programming and mathematics (coeff: 9.251, corr: 0.456)
- `L22/12947` scientific terminology related to healthcare and medical research (coeff: 11.241, corr: 0.440)
- `L23/5752` associations and relationships among scientific variables and observations (coeff: 10.133, corr: 0.497)
- `L24/8188` syntax related to code structure and operations (coeff: 11.861, corr: 0.482)
- `L25/8643` scientific terms and concepts related to biochemistry and cellular processes (coeff: 11.439, corr: **0.545**)

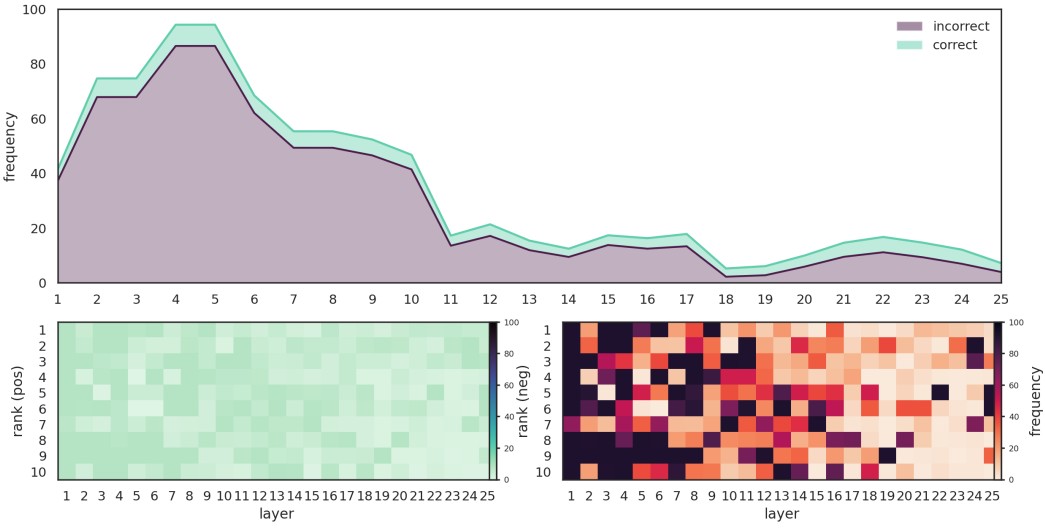

Figure 17: Top correlated features with MMLU-Pro on frequency in each layer of Gemma-2 2B.

**GSM8K**

- `L1/13475` specific quantitative or statistical information (coeff: 9.936, corr: 0.251)
- `L2/2098` references to leadership and management isolation in workplace contexts (coeff: 3.080, corr: 0.180)
- `L3/8338` significant quantities within code snippets, likely indicating important operations or constructs (coeff: 6.302, corr: 0.250)
- `L4/687` HTML tags and attributes related to layout and styling (coeff: 2.037, corr: 0.188)
- `L5/697` terms related to price dynamics and economic relationships (coeff: 6.091, corr: 0.193)
- `L6/13460` references to safety and regulatory issues in automobile contexts (coeff: 9.501, corr: 0.219)
- `L7/9514` structured data or code snippets, potentially relating to geographical regions and associated identifiers (coeff: 1.309, corr: 0.167)
- `L8/2024` names of notable performance venues and cultural institutions (coeff: 14.384, corr: 0.210)

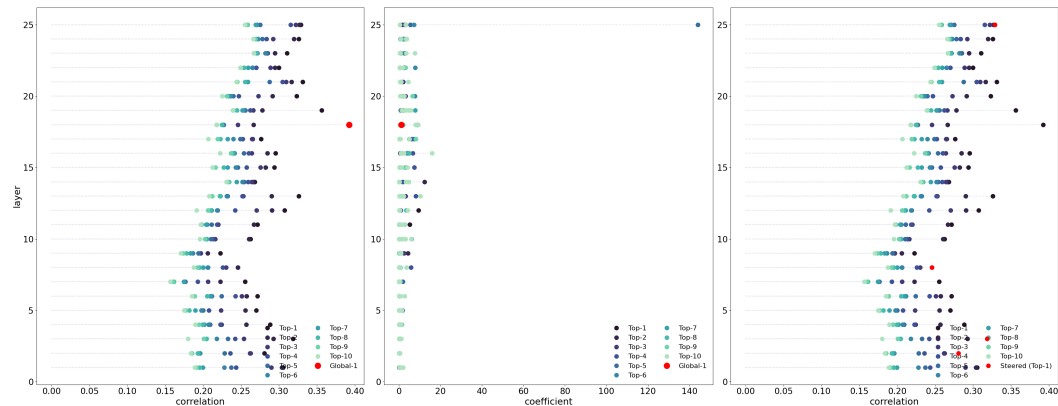

Figure 18: Top correlated features with selected features from CorrSteer-P with GSM8K on coefficient in each layer of Gemma-2 2B.

- L9/15115 discussions related to crime scene investigations and forensic evidence (coeff: 5.074, corr: 0.188)
- L10/2794 elements of conversation or dialogue (coeff: 5.602, corr: 0.188)
- L11/7313 mathematical equations and expressions (coeff: 26.252, corr: 0.176)
- L12/12707 technical or scientific terminology related to systems and processes (coeff: 2.860, corr: 0.245)
- L13/14319 code snippets and their associated structures within documents (coeff: 2.731, corr: 0.253)
- L14/4217 expressions of emotional reactions and feedback (coeff: 3.772, corr: 0.246)
- L15/1685 instances of structured data or messages indicating communication or queries (coeff: 7.282, corr: 0.255)
- L16/14919 instances of unique identifiers or markers in a dataset (coeff: 24.774, corr: 0.223)
- L17/7185 curly braces and structured programming syntax elements (coeff: 6.245, corr: 0.252)
- L18/3732 code syntax elements such as brackets and semicolons (coeff: 4.064, corr: 0.249)
- L19/2015 structures related to function definitions and method calls in programming code (coeff: 8.802, corr: 0.277)
- L20/15616 elements of code structure and syntax in programming contexts (coeff: 4.350, corr: 0.258)
- L21/12547 phrases and words that express confusion or dissatisfaction with situations (coeff: 24.211, corr: 0.251)
- L22/7903 **mathematical notation and symbols used in equations** (coeff: 7.295, corr: 0.313)
- L23/12425 **mathematical expressions and symbols** (coeff: 19.202, corr: 0.294)
- L24/2274 **programming syntax and structure specific to coding languages** (coeff: 10.205, **corr: 0.348**)
- L25/3469 technical aspects related to semiconductor devices and their manufacturing processes (coeff: 23.158, corr: 0.284)

**SimpleQA**

- L1/14904 references to Congress and legislative processes (coeff: 0.263, corr: 0.192)
- L2/1089 terms and concepts related to integrals and the importance of integration in various contexts (coeff: 0.225, corr: 0.228)
- L3/12843 terms related to durability and long-lasting qualities (coeff: 0.219, corr: 0.178)
- L8/10825 punctuation marks and special characters (coeff: 5.194, corr: 0.296)
- L9/9228 punctuation marks, especially periods and quotation marks (coeff: 4.712, corr: 0.323)

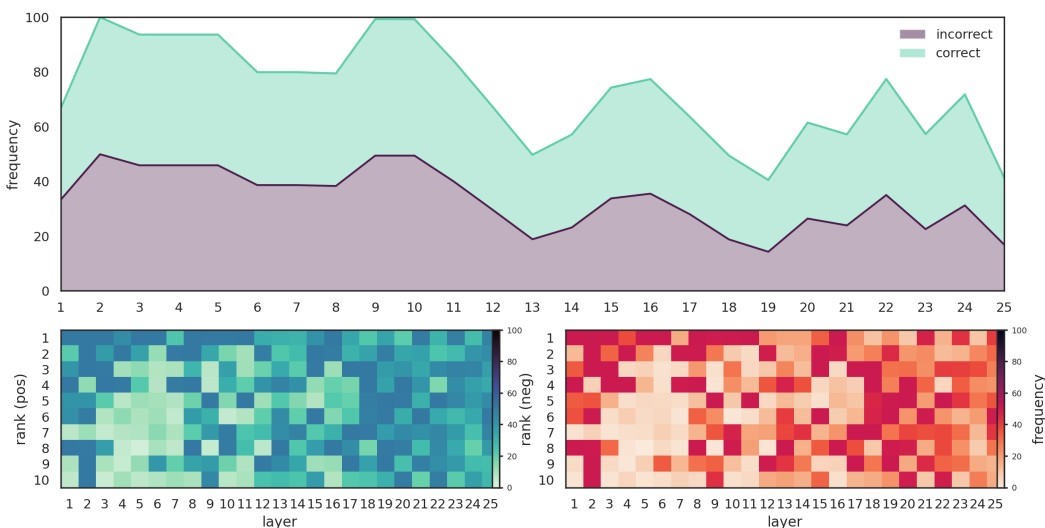

Figure 19: Top correlated features with GSM8K on frequency in each layer of Gemma-2 2B.

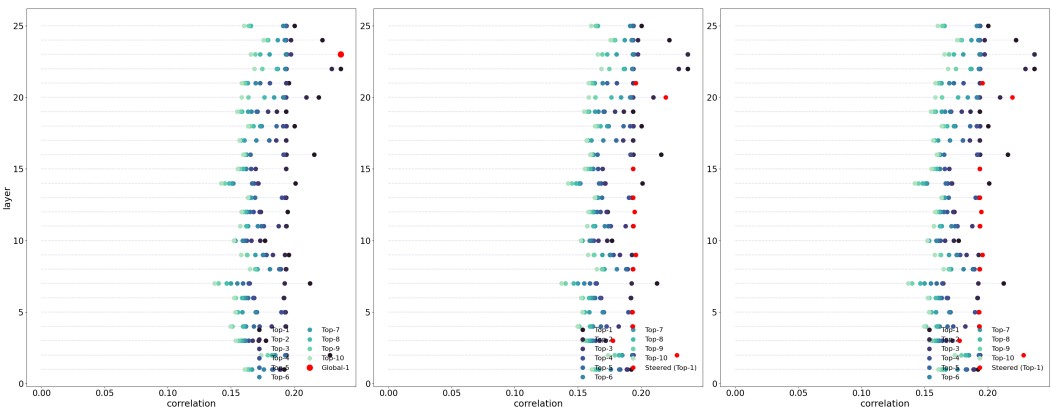

Figure 20: Top correlated features with selected features from CorrSteer-P with SimpleQA on coefficient in each layer of Gemma-2 2B.

- L10/13244 information related to military casualties and incidents (coeff: 2.760, corr: 0.270)
- L11/5734 sections or punctuation that denote lists or explanations (coeff: 4.304, corr: 0.243)
- L12/12342 symbols and mathematical notation related to expressions or equations in mathematical contexts (coeff: 15.373, corr: 0.282)
- L13/10964 mathematical terms and symbols (coeff: 16.622, corr: 0.274)
- L14/7655 structured data, such as XML or JSON formats (coeff: 16.195, corr: 0.275)
- L15/5114 terms related to evaluation and validation processes (coeff: 23.117, corr: 0.248)
- L16/1547 code or programming-related syntax (coeff: 21.527, corr: 0.283)
- L17/10813 references to movies, actors, and significant film industry terms (coeff: 9.662, corr: 0.243)
- L18/8615 legal terminology and concepts related to judicial authority and precedent (coeff: 9.006, corr: 0.282)
- L19/2998 elements related to research findings, including factors, conclusions, and reasoning (coeff: 13.956, corr: 0.245)
- L20/9419 names of individuals and titles (coeff: 10.648, corr: 0.272)
- L21/15170 isolated segments of code or technical content (coeff: 36.804, corr: 0.264)

- L22/11042 punctuation marks that indicate the start or end of lists or key points in a text (coeff: 28.482, corr: 0.294)
- L23/8993 structured API documentation elements and syntax (coeff: 23.447, corr: 0.280)
- L24/4448 terms related to scientific analysis and results reporting (coeff: 16.649, corr: 0.287)
- L25/7968 elements related to health assessments and metrics (coeff: 9.863, corr: 0.307)

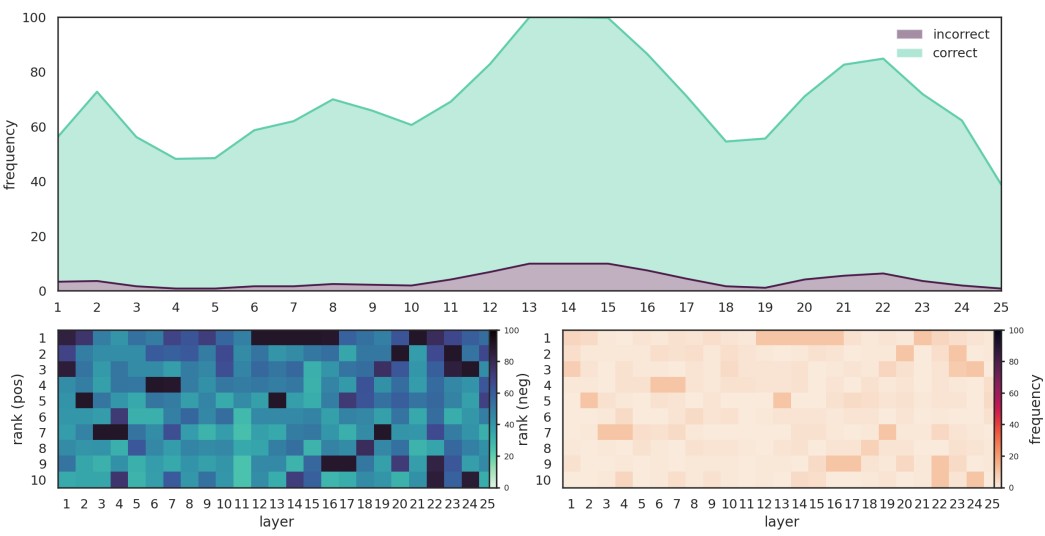

Figure 21: Top correlated features with XSTest on frequency in each layer of Gemma-2 2B.

### A.11.2 LLAMA-3.1-8B

**BBQ (Ambiguous)**

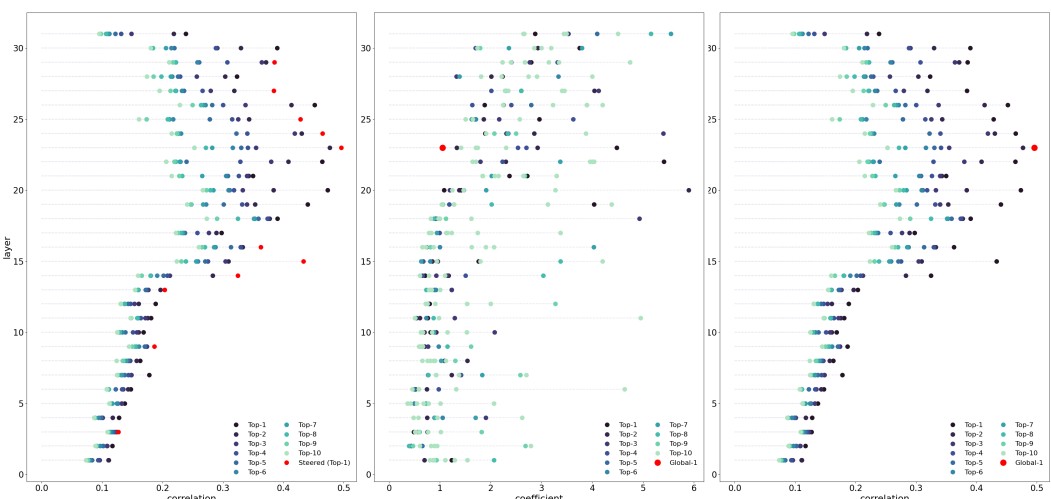

Figure 22: Top correlated features with selected features from CorrSteer-P with BBQ ambig on coefficient in each layer of LLaMA-3.1 8B.

- L1/23207 phrases related to legal or regulatory frameworks (coeff: 0.463, corr: 0.111)
- L2/2680 titles and key information related to television series episodes (coeff: 0.002, corr: 0.117)
- **L3/23846** discussions around societal structures and issues related to mental health and crime (coeff: 0.487, corr: 0.127)
- L4/30896 occurrences of numerical values and references to measurements (coeff: 0.089, corr: 0.128)
- L5/18555 instances of past and present tense verbs, particularly focusing on actions and conditions (coeff: 0.193, corr: 0.137)
- L6/25246 technical terms and code snippets related to software development and programming logic (coeff: 0.277, corr: 0.147)
- L7/11878 specific numerical identifiers and related metadata in technical documents (coeff: 0.365, corr: 0.178)
- L8/4790 keywords related to data structures and programming concepts (coeff: 0.172, corr: 0.163)
- **L9/2700** references to extraterrestrial or paranormal beings and phenomena (coeff: 0.354, corr: 0.187)
- L10/23355 **phrases or constructs that emphasize comparison or simile** (coeff: 0.812, corr: 0.168)
- L11/18132 references to specific books, movies, or artworks (coeff: 0.167, corr: 0.181)
- L12/14096 references to specific locations or settings in various contexts (coeff: 0.084, corr: 0.189)
- **L13/26526** references to error handling in programming (coeff: 0.493, corr: 0.203)
- **L14/13393** statistical percentages and survey data (coeff: 0.192, corr: 0.324)
- **L15/25166** **themes of neutrality and balance in discourse** (coeff: 0.259, corr: 0.433)
- **L16/21816** phrases related to financial or economic assessments (coeff: 0.543, corr: 0.363)
- L17/5782 references to equality and equity in rights and opportunities (coeff: 0.368, corr: 0.298)
- L18/28196 references to knowledge, learning, and understanding in various contexts (coeff: 0.303, corr: 0.390)
- **L19/29460** **discussions about extremes and balance** (coeff: 0.811, corr: 0.440)
- **L20/13319** **expressions of mixed opinions or complex character evaluations** (coeff: 1.413, corr: 0.473)

- L21/8518 references to articles and citations in academic databases (coeff: 2.719, corr: 0.349)
- **L22/28263** **percentages and statistical data concerning opinions or responses** (coeff: 1.024, corr: 0.464)
- **L23/638** formal structures and procedures within organizational contexts (coeff: 1.054, corr: **0.496**)
- **L24/19174** code constructs and control flow keywords related to conditions and returns (coeff: 1.890, corr: 0.465)
- **L25/10753** **expressions of perception or belief in social dynamics** (coeff: 1.147, corr: 0.428)
- **L26/27899** code structure and logical operations involving object hierarchy and data types (coeff: 1.025, corr: 0.452)
- **L27/1765** quantitative data related to project development and financial metrics (coeff: 2.597, corr: 0.384)
- **L28/21019** financial data and statistics related to development projects (coeff: 0.856, corr: 0.323)
- **L29/17998** code snippets related to JavaScript or Java programming functions and structures (coeff: 1.735, corr: 0.385)
- **L30/17084** numerical data related to financial projections and resource development (coeff: 1.308, corr: 0.390)
- **L31/10728** auxiliary verbs and words indicating obligation or possibility (coeff: 1.530, corr: 0.239)

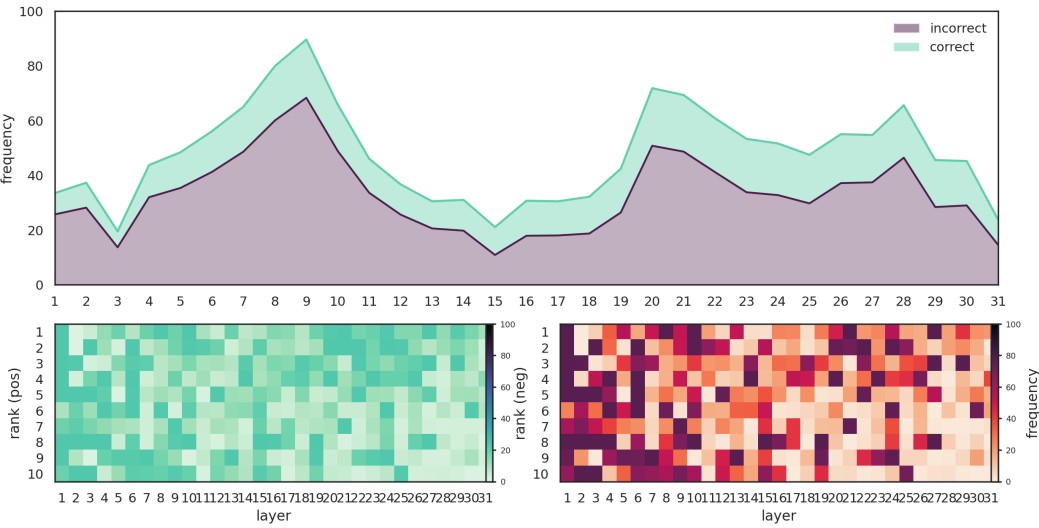

Figure 23: Top correlated features with BBQ ambig on frequency in each layer of LLaMA-3.1 8B.

**BBQ (Disambiguous)**

- **L1/5891** technical terms and references in programming and development contexts (coeff: 0.154, corr: 0.086)
- L2/21865 references to essays, articles, and related writing concepts (coeff: 0.784, corr: 0.084)
- **L3/3413** elements related to user engagement and user-friendly design (coeff: 0.332, corr: 0.100)
- **L4/3712** elements related to programming and computation (coeff: 0.458, corr: 0.086)
- **L5/18066** references to educational administration and school district issues (coeff: 0.229, corr: 0.118)
- L6/28294 references to machine learning models and recommendation systems (coeff: 0.301, corr: 0.119)

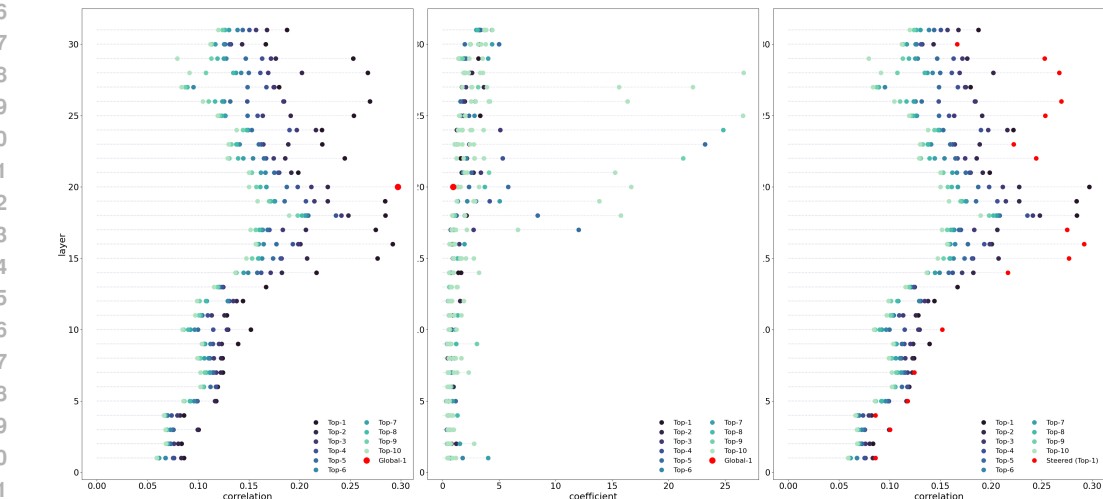

Figure 24: Top correlated features with selected features from CorrSteer-P with BBQ disambig on coefficient in each layer of LLaMA-3.1 8B.

- **L7/7762** specific language constructs related to coordination and organization (coeff: 0.416, corr: 0.124)
- L8/25466 terms related to hierarchical structures or classifications (coeff: 1.032, corr: 0.124)
- L9/5313 key concepts related to project management and planning (coeff: 0.645, corr: 0.139)
- **L10/13407 negative actions and attitudes that hinder interpersonal relationships and community engagement** (coeff: 0.256, corr: 0.152)
- L11/18350 references to institutions and systems regarding public services (coeff: 0.900, corr: 0.128)
- **L12/13336 phrases and concepts related to community and social interactions** (coeff: 0.377, corr: 0.144)
- **L13/15793** negation phrases and words indicating absence or lack (coeff: 0.695, corr: 0.167)
- **L14/31962** details related to physical displacement or movement in a spatial context (coeff: 1.384, corr: 0.217)
- **L15/2128** references to programming elements and constructs (coeff: 0.977, corr: 0.277)
- **L16/6219 code-related syntax and structures within programming languages** (coeff: 0.830, corr: **0.292**)
- **L17/12610** technical terminology related to programming and software development (coeff: 0.706, corr: 0.275)
- L18/16458 HTML tags and structured data elements (coeff: 2.113, corr: 0.285)
- L19/6432 numerical values and the structure of dates or game scores (coeff: 0.909, corr: 0.284)
- L20/28406 tokens related to timestamps, specifically date and time formats (coeff: 0.942, corr: 0.297)
- L21/15538 references to time management techniques and motivational strategies (coeff: 0.388, corr: 0.199)
- **L22/11286** monetary amounts or financial figures (coeff: 0.531, corr: 0.245)
- **L23/30672** phrases involving the concept of answers or responses (coeff: 1.211, corr: 0.222)
- **L24/5888** references to answers or responses in discussions or questions (coeff: 1.152, corr: 0.222)
- **L25/22713** mathematical notations and symbols (coeff: 1.235, corr: 0.253)
- **L26/22133** names of authors and their affiliations in academic contexts (coeff: 1.953, corr: 0.269)
- **L27/12321** structural elements and parameters in programming code or data structures (coeff: 0.539, corr: 0.180)

- **L28/23202 specific numbers and their context within factual statements** (coeff: 1.897, corr: 0.267)
- **L29/3168** keywords related to health and medical terminology (coeff: 3.175, corr: 0.253)
- **L30/22450** terms and phrases related to health and medical conditions (coeff: 3.219, corr: 0.167)
- **L31/18173** procedural commands and technical instructions related to software and settings (coeff: 1.440, corr: 0.188)

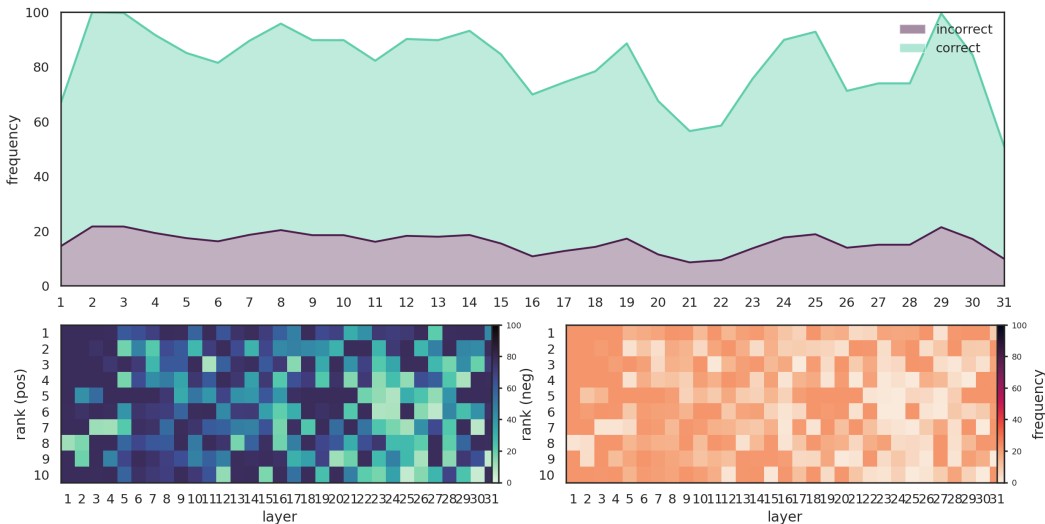

Figure 25: Top correlated features with BBQ disambig on frequency in each layer of LLaMA-3.1 8B.

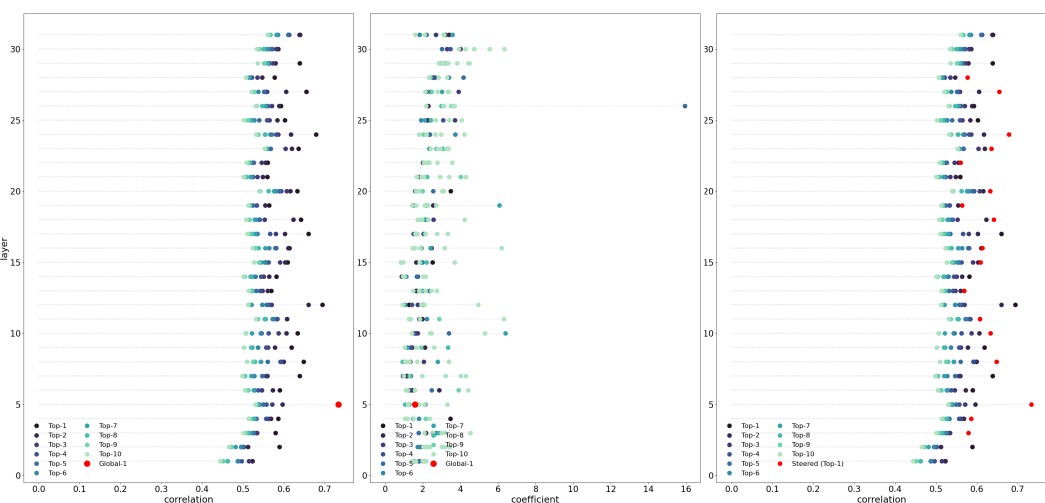

Figure 26: Top correlated features with selected features from CorrSteer-P with HarmBench on coefficient in each layer of LLaMA-3.1 8B.

**HarmBench**

- **L1/15747** repetitive phrases or expressions related to certainty or emphasis (coeff: 0.491, corr: 0.524)
- **L2/25715** references to collective experiences and communal responsibility (coeff: 1.032, corr: 0.590)
- **L3/23621 negations and assertions related to existence and actions** (coeff: 1.116, corr: 0.580)

- **L4/26750 first-person pronouns indicating personal experiences and thoughts** (coeff: 3.468, corr: 0.586)
- **L5/300 instances of political criticism and hypocrisy** (coeff: 1.587, corr: **0.734**)
- **L6/21616 discussions about legality, morality, and the implications of actions in ethical contexts** (coeff: 1.458, corr: 0.590)
- **L7/17622 phrases related to trust and loyalty in political contexts** (coeff: 1.128, corr: 0.639)
- **L8/6508 expressions related to the condemnation of sexual assault and violence** (coeff: 1.322, corr: 0.648)
- **L9/27026 concepts related to limits and responsibilities in relationships and societal interactions** (coeff: 1.425, corr: 0.619)
- **L10/9364 expressions of moral outrage and condemnation regarding social and ethical issues** (coeff: 1.324, corr: 0.633)
- **L11/16561 expressions of personal opinion and moral judgments** (coeff: 1.810, corr: 0.608)
- **L12/5839 strong statements against violence and discrimination** (coeff: 1.271, corr: 0.694)
- **L13/15443** emotional expressions of affection or attachment (coeff: 1.637, corr: 0.569)
- **L14/22046 phrases and sentiments associated with moral judgments and emotional responses** (coeff: 0.750, corr: 0.582)
- **L15/5498** phrases related to environmental and climate impact (coeff: 0.696, corr: 0.609)
- **L16/8375** topics related to stigma and mental health awareness (coeff: 0.938, corr: 0.614)
- **L17/15876 expressions of self-doubt or uncertainty** (coeff: 0.582, corr: 0.660)
- **L18/6210** phrases related to educational support and challenges faced by teachers (coeff: 0.964, corr: 0.641)
- **L19/5854** references to seeking medical advice and guidance (coeff: 1.148, corr: 0.564)
- **L20/11388 elements related to moral and ethical dilemmas** (coeff: 3.490, corr: 0.633)
- **L21/9674 references to racism and social justice issues** (coeff: 0.712, corr: 0.559)
- **L22/4650** expressions of self-awareness and personal growth mixed with skepticism towards collective beliefs (coeff: 2.235, corr: 0.560)
- **L23/28291 phrases discussing social justice and advocacy for marginalized communities** (coeff: 2.165, corr: 0.636)
- **L24/21055** phrases related to self-identity and personal reflection (coeff: 2.357, corr: 0.679)
- **L25/16450 themes of emotional struggle and interpersonal relationships** (coeff: 2.415, corr: 0.602)
- **L26/6648 phrases indicating moral judgment or hypocrisy in political discourse** (coeff: 1.541, corr: 0.593)
- **L27/10654** expressions of emotional conflict and personal reflection (coeff: 1.653, corr: 0.655)
- **L28/522** themes of courage and resilience in writing (coeff: 0.915, corr: 0.578)
- **L29/13883 complex emotional responses and reflections on interpersonal relationships** (coeff: 2.977, corr: 0.639)
- **L30/4588 expressions of emotional needs and desires in relationships** (coeff: 1.480, corr: 0.586)
- **L31/31181 references to familial relationships and memorial details** (coeff: 1.218, corr: 0.639)

**MMLU**

- **L1/4557** specific numeric values and measurements related to instructions or guidelines (coeff: 0.695, corr: 0.094)
- **L2/27893** terms related to technology, specifically graphics processing units (GPUs) and their applications (coeff: 0.348, corr: 0.157)
- **L3/204 terms and concepts related to financial metrics and performance evaluation** (coeff: 1.037, corr: 0.139)
- **L4/23545** questions that lead to detailed inquiries or clarifications (coeff: 1.142, corr: 0.131)
- **L5/17458 terms related to theoretical concepts and methodologies in scientific discussions** (coeff: 0.497, corr: 0.124)

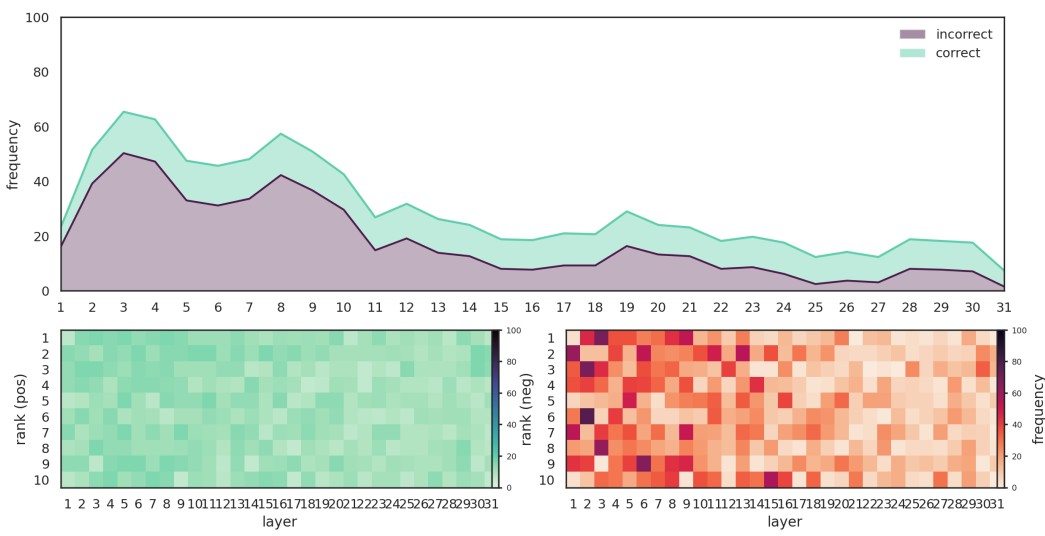

Figure 27: Top correlated features with HarmBench on frequency in each layer of LLaMA-3.1 8B.

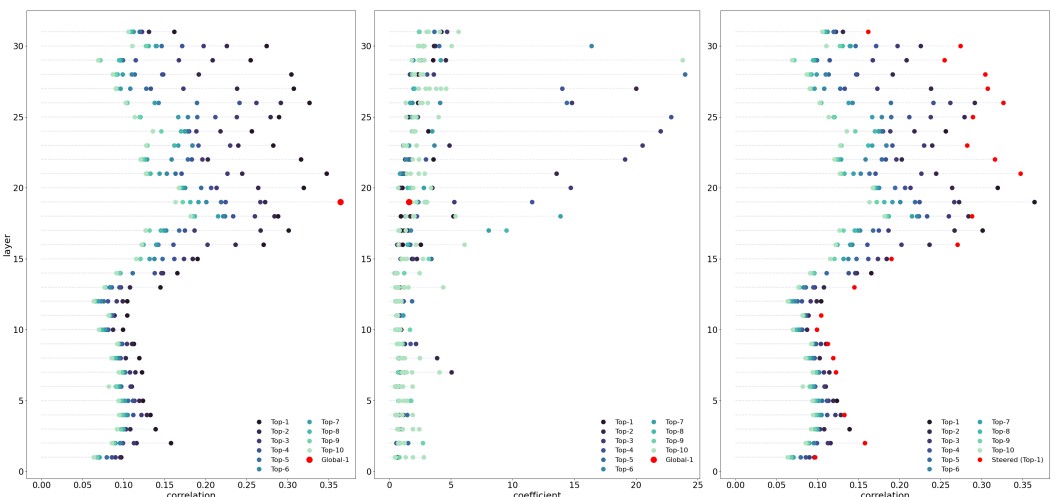

Figure 28: Top correlated features with selected features from CorrSteer-P with MMLU ambig on coefficient in each layer of LLaMA-3.1 8B.

- L6/650 specific identifiers, particularly those related to content or lists (coeff: 0.780, corr: 0.110)
- L7/13659 references to lists, particularly those pertaining to security or classification contexts (coeff: 0.885, corr: 0.118)
- L8/1649 key terms related to organizational assistance and functionality within various contexts (coeff: 0.871, corr: 0.116)
- L9/19730 various forms of interviews and discussions related to current events or cultural topics (coeff: 0.397, corr: 0.108)
- L10/20495 terms related to requirements and definitions within various contexts (coeff: 0.949, corr: 0.099)
- L11/20851 legal and academic terminology related to charges and reports (coeff: 0.897, corr: 0.100)
- L12/26346 specific nouns and proper names related to various contexts (coeff: 0.454, corr: 0.104)

- **L13/551** terms related to medical results and actions taken toward health management (coeff: 0.830, corr: 0.143)
- **L14/11013** phrases indicating relationships between people or entities (coeff: 0.366, corr: 0.165)
- **L15/9446** expressions of passion and enthusiasm in various contexts (coeff: 0.327, corr: 0.195)
- **L16/6219** code-related syntax and structures within programming languages (coeff: 1.094, corr: 0.274)
- **L17/26604** references to programming concepts and structures (coeff: 0.957, corr: 0.301)
- **L18/28750** structured data elements and patterns, possibly related to programming or data analysis (coeff: 0.936, corr: 0.288)
- **L19/6432** numerical values and the structure of dates or game scores (coeff: 1.587, corr: 0.365)
- **L20/28406** tokens related to timestamps, specifically date and time formats (coeff: 1.051, corr: 0.319)
- **L21/15538** references to time management techniques and motivational strategies (coeff: 1.014, corr: **0.347**)
- **L22/11286** monetary amounts or financial figures (coeff: 1.269, corr: 0.322)
- **L23/15096** phrases related to significant life events and milestones (coeff: 1.125, corr: 0.281)
- **L24/18010** references to dates and significant life events (coeff: 1.631, corr: 0.256)
- **L25/22713** mathematical notations and symbols (coeff: 1.209, corr: 0.287)
- **L26/22133** names of authors and their affiliations in academic contexts (coeff: 2.331, corr: 0.331)
- **L27/19268** references to academic qualifications, research, and involvement in educational activities (coeff: 0.826, corr: 0.310)
- **L28/23202** **specific numbers and their context within factual statements** (coeff: 2.318, corr: 0.307)
- **L29/3168** keywords related to health and medical terminology (coeff: 3.545, corr: 0.255)
- **L30/23403** terms associated with uncertainty and error (coeff: 0.986, corr: 0.274)
- **L31/6722** instances of code-related syntax and formatting (coeff: 0.538, corr: 0.159)

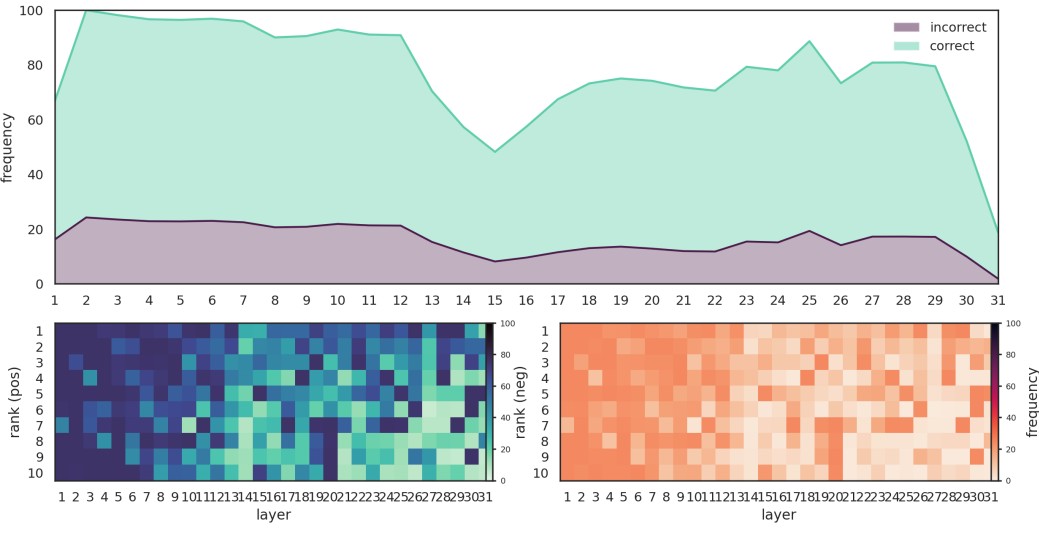

Figure 29: Top correlated features with MMLU on frequency in each layer of LLaMA-3.1 8B.

**MMLU-Pro**

- **L1/2403** specific numeric values and measurements related to instructions or guidelines (coeff: 0.286, corr: 0.216)
- **L2/85** phrases related to service expectations and quality assurance

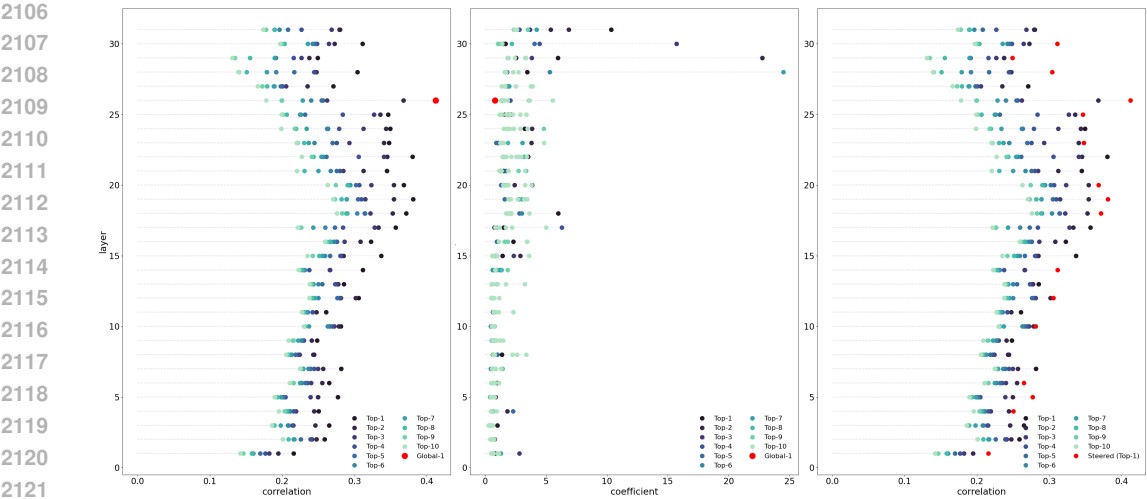

Figure 30: Top correlated features with selected features from CorrSteer-P with MMLU-Pro ambig on coefficient in each layer of LLaMA-3.1 8B.

(coeff: 0.212, corr: 0.259)
- L3/204 terms and concepts related to financial metrics and performance evaluation (coeff: 0.996, corr: 0.265)
- L4/14539 content related to sources and references in articles (coeff: 0.432, corr: 0.250)
- L5/2831 references to urgency and scheduling events (coeff: 0.348, corr: 0.277)
- L6/7784 instances of various relational and transactional terms within context (coeff: 0.153, corr: 0.265)
- L7/22238 references to examples or lists in discussions or reports (coeff: 0.446, corr: 0.282)
- L8/7704 keywords related to television series and their reception (coeff: 0.630, corr: 0.244)
- L9/4007 references to various types of businesses and their classifications (coeff: 0.298, corr: 0.248)
- L10/3783 key phrases and concepts related to business development and investment processes (coeff: 0.454, corr: 0.281)
- L11/7301 components of structured data or content organization (coeff: 0.807, corr: 0.261)
- L12/28750 financial terms and conditions related to trading or commerce (coeff: 0.563, corr: 0.306)
- L13/16587 phrases indicating action or involvement in events or developments (coeff: 0.366, corr: 0.285)
- L14/28135 references to specific geographic locations or entities (coeff: 0.490, corr: 0.312)
- L15/9446 expressions of passion and enthusiasm in various contexts (coeff: 0.425, corr: 0.337)
- **L16/6219 code-related syntax and structures within programming languages (coeff: 0.342, corr: 0.323)**
- **L17/26604references to programming concepts and structures (coeff: 0.469, corr: 0.357)**
- L18/2624 references to criminal activity and associated legal consequences (coeff: 0.478, corr: 0.371)
- **L19/6432 numerical values and the structure of dates or game scores (coeff: 0.966, corr: 0.381)**
- L20/28406 tokens related to timestamps, specifically date and time formats (coeff: 0.628, corr: 0.368)
- L21/15538 references to time management techniques and motivational strategies (coeff: 0.391, corr: 0.345)
- L22/11286 monetary amounts or financial figures (coeff: 0.697, corr: 0.380)
- L23/21146 programming and coding structures, particularly related to network protocols and data handling (coeff: 0.853, corr: 0.348)

- `L24/7967` references to specific locations or addresses (coeff: 0.837, corr: 0.350)
- `L25/16619` instances of authorship and attribution in the text (coeff: 0.864, corr: 0.347)
- **`L26/22133` names of authors and their affiliations in academic contexts(coeff: 0.813, corr: 0.413)**
- **`L27/19268` references to academic qualifications, research, and involvement in educational activities (coeff: 0.318, corr: 0.271)**
- `L28/23202` specific numbers and their context within factual statements (coeff: 1.120, corr: 0.304)
- `L29/12442` patterns related to digital platforms and software updates (coeff: 2.528, corr: 0.249)
- `L30/19427` specific numerical values and statistical data (coeff: 0.374, corr: 0.311)
- `L31/9926` numbers, particularly in relation to financial data and statistics (coeff: 10.348, corr: 0.280)

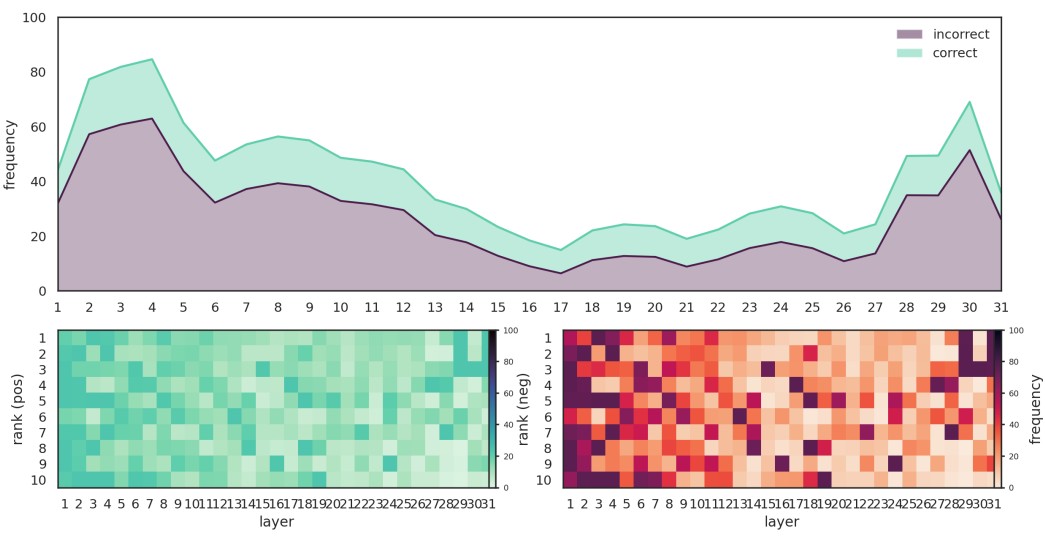

Figure 31: Top correlated features with MMLU-Pro on frequency in each layer of LLaMA-3.1 8B.

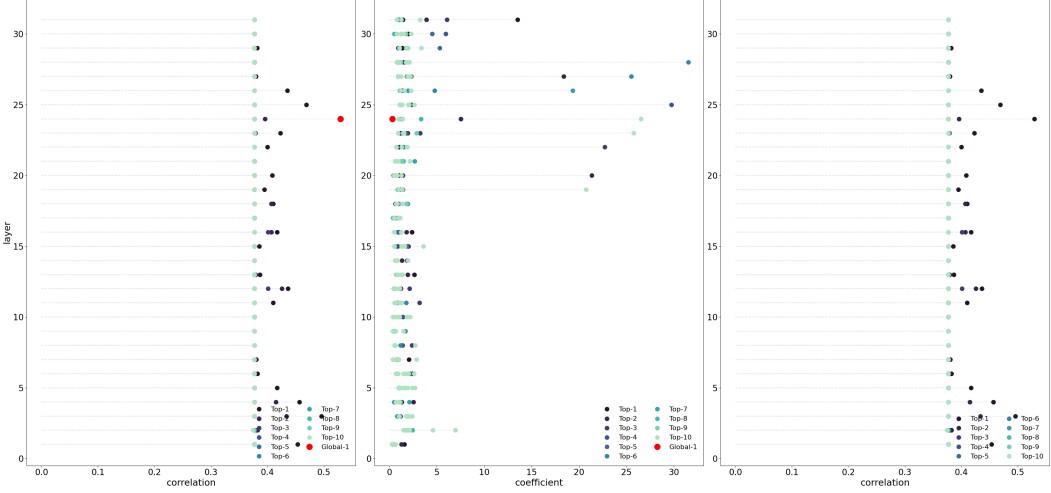

Figure 32: Top correlated features with SimpleQA on frequency in each layer of LLaMA-3.1 8B.

**SimpleQA**

- L1/28160 references to height, specifically focusing on the term "tall" (coeff: 1.580, corr: 0.454)
- L2/16190 references to geographical locations, particularly islands (coeff: 0.148, corr: 0.383)
- L3/24193 references to deserts and desert-related imagery (coeff: 0.541, corr: 0.496)
- L4/25100 references to dumpster rental services and pricing (coeff: 0.205, corr: 0.457)
- L5/15924 the occurrence of the word "in" and its context within the text (coeff: 0.396, corr: 0.418)
- L6/7008 references to artificial entities and technologies (coeff: 2.402, corr: 0.383)
- L7/6257 terms and phrases related to artificial elements or creations (coeff: 2.049, corr: 0.381)
- L8/30264 phrases or terms that indicate suitability or excellence in context (coeff: 0.029, corr: 0.377)
- L9/23784 programming-related keywords and constructs (coeff: 0.089, corr: 0.377)
- L10/30120 phrases that encourage action or reminders related to specific tasks (coeff: 0.057, corr: 0.377)
- L11/962 conjunctions that introduce reasoning or causation (coeff: 0.396, corr: 0.410)
- L12/31391 references to authors and their written works (coeff: 0.472, corr: 0.437)
- **L13/19013 references to biological family classifications (coeff: 2.618, corr: 0.387)**
- L14/12579 references to global outreach and international presence (coeff: 0.077, corr: 0.377)
- **L15/18867 references to biological classifications, specifically family names in taxonomy (coeff: 2.004, corr: 0.386)**
- **L16/22032 biological classifications of species, particularly family and genus names (coeff: 2.364, corr: 0.417)**
- L17/30566 phrases related to ownership or affiliation (coeff: 0.884, corr: 0.377)
- L18/24624 specific terms associated with the media and entertainment industry (coeff: 0.952, corr: 0.410)
- L19/25841 references to personal growth and transformation experiences (coeff: 1.140, corr: 0.395)
- L20/23840 references to legislative districts and redistricting processes (coeff: 0.438, corr: 0.409)
- L21/9851 references to volcanic activity (coeff: 0.258, corr: 0.377)
- L22/20579 references to educational programs and initiatives (coeff: 0.744, corr: 0.400)
- L23/11708 complex arguments and perspectives in academic discourse (coeff: 0.323, corr: 0.423)
- **L24/14877 specific procedural or data-related elements in formal documents (coeff: 0.292, corr: 0.530)**
- L25/18055 words associated with appreciation and commendation (coeff: 0.542, corr: 0.469)
- L26/10617 emotional expressions and relationships in personal narratives (coeff: 0.317, corr: 0.435)
- L27/135 activities related to travel and tourism (coeff: 0.924, corr: 0.380)
- L28/29877 references to the concept of "home." (coeff: 0.964, corr: 0.377)
- L29/4392 references to clothing and dress codes, particularly in relation to gender identity and expression (coeff: 0.410, corr: 0.382)
- L30/22633 public methods in a programming context (coeff: 0.310, corr: 0.377)
- L31/6171 references to artificial intelligence and its related concepts (coeff: 1.429, corr: 0.377)

**XSTest**

- L1/6754 references to studies and publications (coeff: 0.256, corr: 0.367)
- L2/5332 names and characteristics associated with aviation or flight (coeff: 0.276, corr: 0.331)
- L3/16461 terms related to marine life and conservation efforts (coeff: 1.265, corr: 0.394)
- L4/2446 proper nouns and specific entities (coeff: 0.310, corr: 0.334)
- L5/25000 names of notable individuals and places related to historical or cultural significance (coeff: 0.862, corr: 0.354)

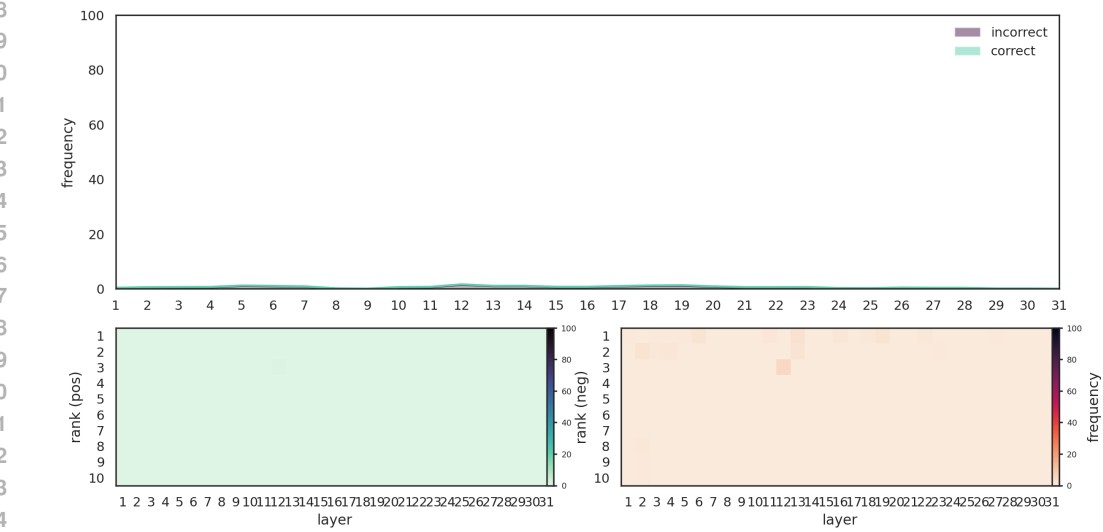

Figure 33: Top correlated features with SimpleQA on frequency in each layer of LLaMA-3.1 8B.

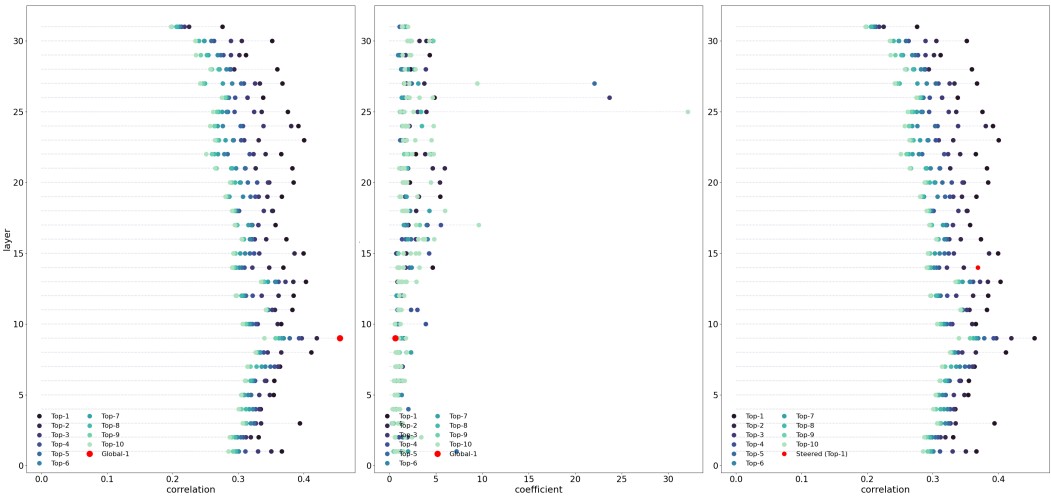

Figure 34: Top correlated features with XSTest on frequency in each layer of LLaMA-3.1 8B.

- L6/10424 information related to personal details and statistics about individuals (coeff: 0.220, corr: 0.355)
- L7/20235 words and phrases associated with measurement or assessment (coeff: 0.784, corr: 0.364)
- L8/22807 concepts related to capital budgeting and investment decision-making (coeff: 0.420, corr: 0.411)
- **L9/16423 references to specific organizations, laws, or conditions related to societal issues (coeff: 0.636, corr: 0.455)**
- L10/11238 phrases related to collaboration and community involvement (coeff: 0.880, corr: 0.365)
- **L11/29172 legal terminology related to civil rights and obligations (coeff: 0.618, corr: 0.383)**
- **L12/19663 negative descriptors or concepts related to cowardice and existence (coeff: 0.735, corr: 0.384)**
- L13/19506 numeric or alphanumeric strings and specific identifiers (coeff: 0.608, corr: 0.403)

- **L14/13505 structured question-answer formats and indicators of a discussion or inquiry (coeff: 4.659, corr: 0.369)**
- **L15/23853 references to female characters and their relationships in narratives (coeff: 0.682, corr: 0.400)**
- L16/1652 names and identifiers related to locations and organizations (coeff: 1.220, corr: 0.373)
- L17/21476 references to influential figures in scientific history and significant concepts from their work (coeff: 2.046, corr: 0.357)
- L18/25543 names and specific references related to individuals, locations, and organizations in a political context (coeff: 0.941, corr: 0.353)
- L19/2102 significant historical events and their impact on society (coeff: 1.691, corr: 0.366)
- L20/21486 various references to awards, accolades, and notable achievements within literary and cinematic contexts (coeff: 2.183, corr: 0.385)
- L21/8477 references to influential figures and their contributions in various contexts (coeff: 2.008, corr: 0.383)
- L22/16870 references to disasters and their impacts (coeff: 2.837, corr: 0.366)
- L23/15524 references to specific events or characters in films (coeff: 1.834, corr: 0.400)
- L24/15231 references to specific events or characters in films (coeff: 1.747, corr: 0.392)
- L25/16855 references to corporate entities and financial transactions (coeff: 0.763, corr: 0.375)
- L26/1578 references to specific individuals or organizations involved in social causes or environmental conservation (coeff: 0.948, corr: 0.338)
- L27/11758 connections to authoritative figures and organizational roles (coeff: 1.300, corr: 0.367)
- L28/425 instances of specific names and organizational references in a text (coeff: 2.291, corr: 0.360)
- L29/17372 terms related to health and illness (coeff: 0.888, corr: 0.312)
- L30/11223 titles and descriptors of programs or services related to community support (coeff: 4.643, corr: 0.352)
- L31/2111 descriptions and features of software products (coeff: 1.614, corr: 0.276)

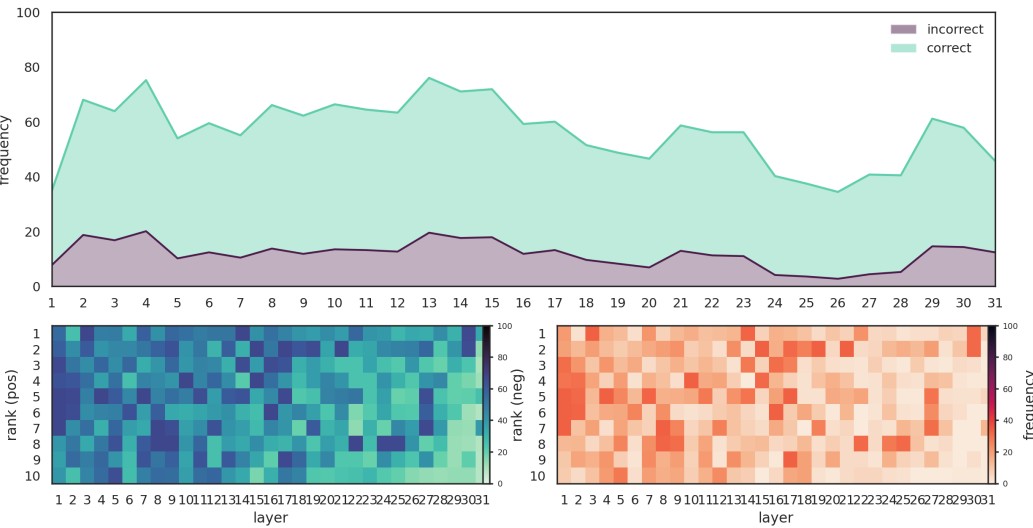

Figure 35: Top correlated features with XSTest on frequency in each layer of LLaMA-3.1 8B.

