# OpenReview forum: "CorrSteer: Generation-Time LLM Steering via Correlated Sparse Autoencoder Features"
_ICLR.cc/2026/Conference — Submitted to ICLR 2026_

### Official Review · Reviewer_kbSg · 2025-10-24

**Soundness:** 3
**Presentation:** 2
**Contribution:** 2
**Rating:** 4
**Confidence:** 3

**Summary:**

This paper proposes Sparse Auto-Encoder (SAE) driven LLM steering during the inference (token generation). The paper argues that task specific steering can be effective if we weight the steering vector based on the linear correlation between SAE features and model outputs across all samples. The paper shows that the proposed approach improves accuracy on a number of benchmarks.

**Strengths:**

+ The paper introduces a simple yet effective method based on SAEs that helps to improve model accuracy on a number of tasks.
+ The paper benchmarks proposed method against a number of baselines and quantifies not only the accuracy on specific tasks but also its unwanted side effects.

**Weaknesses:**

+ The approach sounds interesting but the paper lacks end-to-end description of the algorithm.
Algorithm listing 1 describes the calculation of the coefficients for a given feature i but the end to end algorithm of how and when are these features added is lacking. The authors describe it in the manuscript but the flow of the entire algorithm is unclear.
+ Equation 2 relies only on the SAE features and not on r_i correlation coefficients. What is the exact relationship between the r_i and c_i ?
+ The introductory figure 1 is also somewhat confusing. For example it is unclear how to interpret the normal distribution of the features for prompt 1 and 2. It is unclear why we have 'steering coefficient' subtitle for the intended response and 'positively correlated' subtitle for the unintended prompt response. Overall, it would be good to explain the whole figure and the end-to-end process more clearly.

**Questions:**

+ Are the steering coefficients computed on the same dataset as the evaluation on table 5 ?
+ A clarification question regarding this statement:
`
We apply steering exclusively at token positions where the corresponding features were extracted, rather than uniformly across all tokens (Soo et al., 2025) or restricted to the final token`

 When we say that `we apply steering exclusively at token positions where the corresponding features were extracted,` what does it mean `where the corresponding features are extracted` ? Does extraction mean generation here ? During the inference process we don’t know what token will be generated but we have top candidates.

+ To which exact positions is c_i*W_dec[:,i] steering vector  being added  ?

**Details Of Ethics Concerns:**

N / A

---

> ### Author Response · Authors · 2025-11-20
> **Official Comment by Authors**
>
> We thank the reviewer for the constructive feedback. Below we address each concern.
>
> ## 1. End-to-end algorithm flow and when steering vectors are applied
>
> **Reviewer Concern:** Algorithm 1 describes coefficient calculation for a given feature i, but which positions is $c_i \cdot \mathbf{W}_{\text{dec}}[:, i]$ steering vector being added?
>
> **Author Response:** Selected features are applied only to the residual stream of generated tokens, excluding the prompt tokens. We apply steering at the same positions where correlation activations were computed, ensuring consistency between measurement and steering:
>
>
> $$
> t < n \\Rightarrow\\; \mathbf{x_t}' = \mathbf{x_t}
> $$
>
> $$
> t \ge n \\;\Rightarrow\\; \mathbf{x_t}' = \mathbf{x_t} + \sum_{i \in \mathcal{F}} c_i \cdot \mathbf{W}_{\text{dec}}[:, i]
> $$
>
>
>
> where $n$ is the prompt length, $\mathcal{F}$ denotes the set of selected features, and $t$ is the token position.
>
> The complete pipeline:
> 1. **Feature Selection:** Extract SAE activations from newly generated tokens → Compute correlations with outcomes → Select top positively correlated features (CorrSteer-S/A/P)
> 2. **Coefficient Calculation:** Compute $c_i$ as average activation over positive samples for each selected feature $i$
> 3. **Steering (Inference):** For each newly generated token, add $\sum_{i \in \mathcal{F}} c_i \cdot \mathbf{W}_{\text{dec}}[:, i]$ to the residual stream
>
>
>
> ## 2. Relationship between correlation $r_i$ and steering coefficient $c_i$
>
> **Reviewer Concern:** Equation 2 relies only on SAE features and not on $r_i$ correlation coefficients. What is the exact relationship?
>
> **Author Response:** The two coefficients play independent roles.
> - **$r_i$** (feature **selection**): identifies which features correlate with task outcomes.
> - **$c_i$** (steering **magnitude**): determines how much to steer by averaging positive-sample activations.
>
> We empirically tested coupled approaches (e.g., $c_i \times r_i$) but found that the independent calculation performs better and aligns with SAE's linear architecture and the Linear Representation Hypothesis.
> This coefficient calculation enables full pipeline automation without manual tuning.
>
> ## 3. Figure 1 clarity
>
> **Reviewer Concern:** Figure 1 is confusing regarding the normal distributions and subtitle labels.
>
> **Author Response:** We have expanded Figure 1's caption to clarify: red distributions show feature activations for unintended outputs, blue distributions show feature activations for intended outputs. Steering coefficients are computed as the center (mean) of the blue distributions.
>
> ## 4. Training and evaluation data
>
> **Question:** Are the steering coefficients computed on the same dataset as the evaluation?
>
> **Response:** Yes. In Table 5, the steering-coefficient extraction step is fixed to isolate the effect of the pooling method.
> The pooling variants differ solely in how correlations are computed (max, mean, all-token), while the resulting steering coefficients remain identical across all variants.
> All methods use the same train/validation/test split with identical random seeds.
> Steering coefficients extracted from the training split are used unchanged during validation and test evaluation.

---

> ### Comment · Reviewer_kbSg · 2025-11-27
>
> Thank you very much for the detailed response!
>
> **Training and evaluation data**
> 1. If I understand correctly, each new token during the inference time is assigned a score based on the training / validation selection. Doesn’t context matter here ? Especially for generic tokens the context might be crucial. How do you account for the context? E.g. the token `slow` might have different meanings in different contexts but it looks like we are assigning the same score to all generated `slow` tokens ?
>
> **Relationship between correlation $r_i$ and steering coefficient $c_i$**
>
> I think that what is confusing in the paper is that $r_i$ notation on line 126 is not used later. It seems that it needs to be incorporated into line 140. If there was an algorithm listing for examples we could see the connection much better.
>
> **Figure 1**
> It’s still unclear why blue distribution has `steering coefficient` and  the red one `Positively Correlation` titles. It is unclear from the plot what is correlated with what.
> It seems that the authors mean that the red points that are positively correlated with blue points ?

---

> > ### Author Response · Authors · 2025-11-27
> > **Official Comment by Authors**
> >
> > We thank the reviewer for the follow-up questions. We address each below.
> >
> > ## 1. How does context information factor into our method?
> >
> > Context is explicitly accounted for through activations, not through tokens directly. Crucially, we do not correlate individual tokens with correctness; instead, we correlate **sequence-level pooled activations** with **sequence-level correctness**.
> >
> > Specifically, the correctness label is determined at the sequence level based on the benchmark task. We then extract SAE feature activations only from generated tokens, aggregate them via max-pooling across the sequence, and compute correlations between these pooled activations and sequence-level correctness.
> >
> > This approach works because, as the reviewer correctly notes, the same token "slow" can appear in different contexts. However, after the first layer, the attention mechanism mixes each token's representation with previous context, creating hidden states that depend on the full context, not just the individual token. Therefore, even when the same token appears, different SAE features activate depending on the surrounding context. The correlation computation then operates at the sequence level, capturing which features (reflecting both token and context across the entire generation) consistently activate when the model produces correct outputs.
> >
> > ## 2. Relationship between correlation $r_i$ and steering coefficient $c_i$
> >
> > We have added Appendix A.2 "Formal Definition of CorrSteer Variants" that explicitly connects $r_i^\ell$ (line 126) to the steering equation (line 140). The relationship is:
> >
> > Correlation $r_i^\ell$ determines **feature selection** (which features enter $\mathcal{F}$):
> > - **CorrSteer-S**: $\mathcal{F_S} = \\{\text{arg max}_{(\ell,i)} r_i^\ell : r_i^\ell \gt 0\\}$
> > - **CorrSteer-A**: $\mathcal{F_A} = \\{(\ell, i) : i = \text{arg max}_j r_j^\ell, r_i^\ell \gt 0, \forall \ell\\}$
> > - **CorrSteer-P**: $\mathcal{F_P} = \\{(\ell, i) \in \mathcal{F_A} : \text{Acc}^\text{val}(\\{(\ell,i)\\}) \gt \text{Acc}^\text{val}(\emptyset)\\}$
> >
> > Steering coefficient $c_i^\ell$ determines **steering magnitude** for selected features:
> > $$c_i^\ell = \frac{1}{|\{j : y_j \gt 0\}|} \sum_{j : y_j \gt 0} z_{i,j}^\ell$$
> >
> > Applied at inference:
> > $$\tilde{\mathbf{x_t^\ell}} = \mathbf{x_t^\ell} + \sum_{(\ell',i) \in \mathcal{F}, \ell'=\ell} c_i^\ell \cdot \mathbf{W_\text{dec}^\ell[:, i]}$$
> >
> > **Example:** Layer 10, feature 532 has $r_{532}^{10} = 0.65$ (highest). CorrSteer-S selects it into $\mathcal{F_S}$, computes $c_{532}^{10}$ as mean activation over positive samples, and adds $c_{532}^{10} \cdot \mathbf{W_\text{dec}^{10}[:, 532]}$ to each generated token's residual stream at layer 10.
> >
> > ## 3. Figure 1 clarity
> >
> > **What is correlated with what:** The correlation is between **feature activation values** (x-axis) and **correctness** (y-axis: 0 for incorrect, 1 for correct), not between red and blue points.
> >
> > - **Red clusters (y=0)**: Feature activations when model produces incorrect outputs
> > - **Blue clusters (y=1)**: Feature activations when model produces correct outputs
> > - **Positive correlation**: Higher feature activation values are correlated with correct outputs (y=1), lower values with incorrect outputs (y=0)
> >
> > The "Steering Coefficient" label at the blue distribution mean shows where we extract $c_i$ (the average activation over correct samples). We have repositioned the "Positively Correlated" label along the correlation trend to clarify that it refers to the relationship between activation values and correctness, not a label above the red points.
> >
> > ---
> >
> > We hope these clarifications address your concerns. If there are any remaining questions, please let us know. Otherwise, we would greatly appreciate if you would consider increasing your score given the improvements made to the paper.

---

### Official Review · Reviewer_pQu7 · 2025-10-27

**Soundness:** 3
**Presentation:** 2
**Contribution:** 2
**Rating:** 4
**Confidence:** 4

**Summary:**

The paper introduces CorrSteer, a method for steering LLMs through correlation-based selection of interpretable features from SAEs. CorrSteer addresses limitations of prior SAE-based steering approaches by leveraging generation-time activations to identify task-relevant features and determine steering coefficients, eliminating the need for contrastive datasets or large activation storage. The authors demonstrate CorrSteer's effectiveness across a range of benchmarks, including question answering, bias mitigation, and safety, while also providing insights into the semantic coherence of the selected features.

**Strengths:**

1. The authors provide a thorough analysis of the method's performance, including side effect quantification and ablation studies.
2. The method is evaluated on tasks from many different domains and question formats.

**Weaknesses:**

1. According to Table 2, the improvement of SER over FT in the MMLU-Pro and GSM8k tasks appears limited. Additionally, the claim that CorrSteer-P is the most balanced strategy might be somewhat overstated. It would be helpful to include more discussion on how and why different selection strategies influence SER performance.

2. Regarding transferability, I am concerned that the correlation-based method might be hard to balance between selecting features relevant to the specific task and those reflecting the underlying skill of the domain. For instance, as mentioned in Lines 452–456, the strong transfer performance in MMLU might be due to the method capturing general reasoning ability. Could it be that domain-specific features are relatively fewer in this case, leading to a higher proportion of general ability features being selected when applying the threshold? Similarly, the BBQ results show a high coefficient for “L24/16355 definitions and mathematical notation in text (coeff: 39.910)” for Gemma, which may possibly indicate overlap in the domain in the dataset rather than reasoning skills. A more detailed discussion on how to balance general versus skill-specific feature selection would strengthen the analysis.

3. The results and discussion section could be further improved by expanding the discussion and providing deeper insights into these observations.

**Questions:**

1. I would like to clarify what is meant in Lines 449–450: “These results suggest that task-specific semantic features contribute more to accuracy than general recognition features.” I would appreciate a more detailed discussion on this point.

2. I wonder whether different tasks might also influence the choice of strategy, given that the concept spaces could vary and the distribution of features across models may differ accordingly.

3. I noticed that the scales of the coefficients and correlation scores for Gemma and Llama differ considerably. Could you please elaborate on the possible reasons for this discrepancy?

4. It might be better to move some key figures and tables to the main text. For example, Tables 5 and 8 are frequently referenced but are currently placed in the appendix, while Figure 4 and Table 2 present similar results and are often discussed together. Including these in the main section could make the presentation clearer.

---

> ### Author Response · Authors · 2025-11-20
> **Official Comment by Authors**
>
> We thank the reviewer for the constructive feedback. We have moved Tables 5 and 8 (pooling and negative feature ablations) from the appendix to the main text as suggested. Below we address each concern.
>
> ## 1. Balancing general vs. task-specific features
>
> **Reviewer Concern:** Correlation-based selection may struggle to distinguish domain-specific features from general reasoning features. The BBQ example showing "mathematical notation" features may indicate dataset overlap rather than reasoning patterns.
>
> **Author Response:** This is an insightful observation. CorrSteer intentionally does not distinguish between general and task-specific features during selection. Instead, it correlates activations with task correctness, allowing the method to automatically capture whichever features improve performance.
>
> Regarding your question about feature proportions: the relative number of general vs. task-specific features naturally varies by task and dataset. For MMLU, both types coexist (some features represent domain knowledge while others capture structural patterns). The threshold determines how many features are selected, but not their type distribution, which emerges from what correlates with correctness.
>
> In practice, we observe:
> - **MMLU/MMLU-Pro**: Selected features include domain-specific knowledge (scientific terminology, medical terms, mathematical notation) and structural elements (statistical symbols, academic formatting) that collectively support question-answering.
> - **BBQ/HarmBench**: Features include bias-related lexical cues and context-specific patterns, which CorrSteer-P effectively filters through validation-based pruning.
>
> We acknowledge that neither correlation threshold nor validation pruning explicitly distinguishes general from task-specific features (both select based on task correctness). However, the transferability analysis (Table 2) demonstrates this yields a natural balance: MMLU features transfer moderately to BBQ Ambig (64.01% vs 59.10% non-steered), indicating some generalizable patterns are captured, while task-specific features outperform transferred ones (76.53% on BBQ Disambig), confirming both types contribute without requiring explicit separation.
>
> **On Lines 449–450 clarification:**
> We revised the wording to:
>
> > "Task-specific semantic features contribute more to accuracy than meta-cognitive recognition features (e.g., choice-format markers, preference expressions)."
>
> We now provide concrete examples of negatively correlated features that represent meta-cognitive patterns rather than domain knowledge:
>
> - L17/9134: choice-related phrases and expressions of preference (coeff: 2.379, corr: -0.451)
> - L19/15745: decision-making and choice expressions in social contexts (coeff: 9.740, corr: -0.464)
>
> These features represent surface-format or instructional recognition patterns rather than task-solving competence.
>
> ---
>
> ## 2. Strategy Choice Clarification
>
> **Reviewer Concern:** SER improvements over FT appear limited for some tasks, and the claim that CorrSteer-P is the “most balanced” strategy needs further justification.
>
> **Author Response:** We have expanded Table 2 (now Table 4) to report SER for all three strategies with multiple seeds, providing more comprehensive statistics.
> The results show:
> - **CorrSteer-S** minimizes intervention scope but is sensitive to single-layer SAE quality
> - **CorrSteer-A** maximizes accuracy but introduces more intervention points, increasing side effect risk
> - **CorrSteer-P** balances coverage and precision, and in our experiments reduces spurious activations (lower SER) on several safety-sensitive tasks
>
> In our experiments, CorrSteer-P typically maintains accuracy close to CorrSteer-A while achieving SER comparable to CorrSteer-S, so we present it as a conservative middle-ground option and report all three variants since the best choice still depends on the task.
>
> ---
>
> ## 3. Coefficient and Correlation Scale Discrepancies (Gemma vs LLaMA)
>
> **Reviewer Concern:** Why do coefficients and correlation scales differ between the two models?
>
> **Author Response:** We have added a detailed explanation in Appendix A.9. Two primary causes explain these differences:
>
> **1. SAE architecture difference:**
> - LLaMA-Scope uses **TopK SAE**, which enforces fixed sparsity through top-k selection.
> - Gemma-Scope uses **JumpReLU SAE** with adaptive thresholding.
>
> **2. Model and SAE capacity differences:**
> The models differ in base model size (2B vs 8B parameters) and SAE dictionary capacity (16K vs 32K features). These architectural and capacity differences contribute to the observed scale variations.
>
> Examining layer-wise correlation patterns (Appendix A.10), we observe that correlation strength generally increases with layer depth for most tasks (e.g., Gemma MMLU: 0.140 → 0.336, LLaMA BBQ: 0.086 → 0.297).

---

> > ### Comment · Reviewer_pQu7 · 2025-11-24
> >
> > Thank you for clarifying my concern. I have raised my score.

---

### Official Review · Reviewer_pRxr · 2025-11-01

**Soundness:** 2
**Presentation:** 3
**Contribution:** 2
**Rating:** 2
**Confidence:** 3

**Summary:**

This paper presents CorrSteer, a family of methods for selecting features from a dictionary to steer towards some positive outcome. These features have coefficients that are precomputed using positive instances of training data. Then the features are used to steer the model with these coefficients. The different variants have different procedures for selecting which features are being steered for. The method is clearly explained and the paper is well written. However, I am concerned about the baseline evaluation results.

The claim that we shouldn't have to consider difference in means steering as a direct competitor with CorrSteer:

"Head-to-head comparison with CAA (Rimsky et al., 2024), DSG (Muhamed et al., 2025), or SPARE (Zhao et al., 2025) is not directly applicable since these methods require contrastive datasets rather than generation-time features"

Seems only somewhat true. Corrsteer requires precomputing **coefficients** for the features that are steered with using positive instances of data. While this isn't a contrastive dataset, it definitely requires going through a dataset to compute information used to steer. And CAA also requires going through a dataset to compute a steer. I think that comparing CAA using a positive dataset and some other neutral dataset, e.g., the pile or wikipedia, would be a perfectly fair comparison with this method, i.e., adapting contrastive methods to a positive only regime. That being said, the contrastive datasets for these tasks are quite east to compute, so the positive dataset only regime might be better highlighted on a task where negative data is harder to get.

I also am concerned about using the coefficients for the CorrSteer to steer the CAA and other baselines. You should do a separate steering coefficient calculation for each of the other baseline methods that is separate from what you did for CorrSteer. Baseline methods should generally have entirely different parameter tuning procedures.

Finally, I would also like to see the SER results in table for the other baseline methods from table 1, as they continue to be relevant comparison points.

As long as this method requires iterating through a dataset to compute the coefficients, I think it needs to be compared against the standard baselines for steering that iterate through similar datasets. On all datasets your method is within margin of error of the SPARE method in performance, and other baselines also reach the performance. Generally, it feels to me like the benefit of this method would all come down to the fact that it uses only positive instances, but if that is the case the paper should have identified experimental settings where positive instances are inherently cheaper to acquire and negative instances can't be substituted with neutral text. Then, the experimental results would show a delta between this method and existing methods.

I'm open to being convinced otherwise by rebuttal, other reviewers, or further experimental results and raising my score.

Missing citations:  https://arxiv.org/pdf/2310.06824, https://arxiv.org/abs/2308.10248, https://arxiv.org/abs/2205.05124

**Strengths:**

see summary

**Weaknesses:**

see summary

**Questions:**

see summary

---

> ### Author Response · Authors · 2025-11-20
> **Official Comment by Authors**
>
> We thank the reviewer for the constructive feedback. We have added the missing citations suggested by the reviewer. Below, we provide clarifications and additional experimental results to address each concern.
>
> ---
>
> ## 1. Comparison with Contrastive Methods (CAA, SPARE, DSG)
>
> **Reviewer Concern:** CorrSteer also iterates through data, similar to contrastive methods. Why not compare against CAA using positive and neutral samples?
>
> **Author Response:** We agree and have revised the manuscript to clarify that contrastive methods can be adapted to our setting. We softened our earlier statement and explicitly compared CorrSteer to adapted versions of CAA, SPARE, and DSG, all implemented using the same benchmark datasets and generation-time SAE activations.
>
> CorrSteer differs in *how* activation statistics are used:
> - CorrSteer computes **correlations** between sparse SAE feature activations and correctness, using only positive evidence.
> - Contrastive methods compute **activation differences** (correct − incorrect), which often become unstable in sparse SAE space due to zero or sign-inverted activations for incorrect samples.
>
> Both approaches use the same correctness labels to identify positive and negative instances, so data acquisition cost is equivalent. The key difference lies in how these instances are used for feature selection and coefficient computation.
>
> **On positive + neutral CAA:**
> Using positive + neutral text is a valid extension. However, this would require adapting CAA's coefficient formulation, which relies on correct–incorrect activation differences.
>
> All comparisons and position updates are reflected in the latest paper version.
>
> ---
>
> ## 2. Coefficient Calculation for Baselines
>
> **Concern:** Each baseline should ideally use its own coefficient computation.
>
> **Response:** We explicitly decompose steering into two orthogonal components:
>
> - **$r_i$ - feature selection:** SPARE (MI), DSG (Fisher), CorrSteer (correlation);
> - **$c_i$ - coefficient magnitude:** CorrSteer (positive-activation average), CAA (correct−incorrect difference).
>
> SPARE and DSG define feature-selection criteria but do not, to our knowledge, specify a data-driven per-feature coefficient rule for sparse SAE latents in our steering setting. For a clean comparison of selection effectiveness, we therefore evaluate them under CorrSteer’s coefficient rule (positive-activation average), so that differences in performance primarily reflect feature selection rather than coefficient scaling.
>
> CAA, however, defines a native coefficient scheme; we used it while matching feature subsets to isolate coefficient differences.
>
> We also tested coupled coefficient-selection schemes (e.g., $c_i \times r_i$, MI-weighted scaling), but found they lower performance while raising side effect ratio.
>
> Additionally, we analyzed layer-wise correlation patterns and found that correlation strength increases with layer depth (e.g., LLaMA BBQ: 0.086 → 0.297). This analysis is detailed in Appendix A.10.
>
> ---
>
> ## 3. Additional SER Experiments
>
> **Concern:** SER should be reported for all baseline methods.
>
> **Response:** We expanded Table 2 (now Table 4) with SER for all baseline methods.
> CorrSteer-A achieves lower SER than CAA (1.00 on GSM8K, HarmBench) and Fisher (0.42 on MMLU) while remaining competitive with MI (SPARE).
> Side effect patterns also vary with generation length: single-token tasks (MMLU, BBQ) show lower SER, while multi-token generation tasks accumulate more side effects over longer horizons.

---

> > ### Author Response · Authors · 2025-11-27
> > **Official Comment by Authors**
> >
> > Dear Reviewer,
> >
> > We have addressed your concerns with additional experiments and clarifications:
> > - Added SER results for all baseline methods (now Table 4)
> > - Clarified that CorrSteer and contrastive methods use equivalent data (both require correctness labels)
> > - Provided detailed coefficient calculation justification for each baseline and formal variant definitions (Appendix A.2)
> >
> > We would greatly appreciate your feedback on our response. If any concerns remain, we are happy to discuss further. Otherwise, we would be grateful if you would consider updating your score given these improvements.

---

### Official Review · Reviewer_igPZ · 2025-11-04

**Soundness:** 2
**Presentation:** 3
**Contribution:** 2
**Rating:** 4
**Confidence:** 5

**Summary:**

This paper introduces a method for selecting SAE features which are useful for steering by ranking their correlations with labels on some dataset, also with built-in computation of the steering coefficient. There are three ablations for selecting the features and two ablations on how the coefficient is computed. Results-wise, this approach matches performance with full finetuning on several tasks and some of the selected features seem broadly related to the task at hand. Overall the approach is a simple combination of existing ideas in the literature with some interesting results.

**Strengths:**

- The evaluations are comprehensive, covering a few models and several tasks. The analysis presented in the paper (particularly the appendices) does a good job of conveying the results and helping the reader understand the value of the method.

**Weaknesses:**

- My understanding is that the method is not framed as a PEFT but rather a steering vector technique. This makes sense since the method seems to lag behind finetuning performance by a bit in many cases. However, the datasets studied are all standard PEFT benchmarks and not more concept-specific benchmarks for comparing steering vectors, e.g. [AxBench](https://arxiv.org/abs/2501.17148).
- The baselines compared against and the datasets benchmarked on are not adequate for a paper about steering vectors and SAEs. Some methods that need to be compared against are [SAE-TS](https://arxiv.org/abs/2411.02193) which uses SAEs to learn good steering vectors, simple baselines like linear probes, and perhaps more recent approaches like [RePS](https://arxiv.org/abs/2505.20809).
- There are three variants of the method presented in each table. I worry that this encourages overfitting by essentially allowing max @ 3 scoring on the leaderboard; just by chance, one of the variants slightly outperforms the other but the relative ordering of the variants performance-wise is not clear. Generally, scores seem within the standard of deviation shown; due to all this I'm not at all convinced that this technique is actually better, in a statistically significant sense, than the baselines.
- In general I think this paper would be better if it focused on this approach as being an interpretability technique rather than a genuine performance-booster for model finetuning (since the latter is not convincing). The appendices showing which features are highly-correlated with different tasks was more interesting and novel than much of the main text. It would be interesting to compare benchmarks and models by which features they use.

**Questions:**

- Why is algorithm 1 so prominent in the main text? It seems more suitable for the appendix (when discussing system optimisations/implementation etc.) since online computation of correlation is hardly a novel contribution, or even a primary one in this work, unless I am mistaken.
- Why Pearson correlation over other classification metrics (e.g. AUROC)? The labels are discrete for each of the tasks right? I may have misunderstood why $y_i$ represents in the explanation here.

---

> ### Author Response · Authors · 2025-11-20
> **Official Comment by Authors**
>
> We thank the reviewer for the constructive feedback. Below we address each concern.
>
> ## 1. CorrSteer positioning and baseline comparisons
>
> **Reviewer Concern:** Method is framed as steering rather than PEFT.
>
> **Author Response:** CorrSteer is a steering technique, not PEFT. PEFT methods modify weights, CorrSteer adds static residual vectors at inference time without weight updates or trainable parameters, yielding lower inference overhead.
>
> **Why standard benchmarks?** CorrSteer targets task-level steering (e.g., improving QA accuracy and reducing bias) rather than concept-level steering; it generalizes SAE-based concept steering to generation-time task steering by applying correlated features.
>
> ## 2. Three variants and statistical significance
>
> **Reviewer Concern:** Three variants (S/A/P) enable "max@3" selection; unclear statistical significance.
>
> **Author Response:** The variants represent **ablations** of feature selection strategies: CorrSteer-S (single global feature), CorrSteer-A (all-layer), and CorrSteer-P (validation-pruned). For comparison with other SAE steering methods under the same multi-layer setting, we report CorrSteer-A. We also added multi-seed statistics and baseline SER results to strengthen statistical robustness. These clarifications are reflected in Section 5.1 (Comparison with Baselines).
>
>
> ## 3. Pearson correlation vs. AUROC
>
> **Reviewer Concern:** Why Pearson correlation for discrete labels instead of AUROC?
>
> **Author Response:** We agree that the choice of metric should be clearly justified. AUROC has known limitations under class imbalance, as shown in [Oikarinen et al. (2025)](https://openreview.net/forum?id=8unyWZ14mf), Fig. 3. Additionally, it relies only on activation ranks rather than magnitudes, which may discard important information from SAE activations when feature intensity carries semantic meaning.
>
> For these reasons, we follow the NeuralEval framework ([Oikarinen et al., 2025](https://openreview.net/forum?id=8unyWZ14mf)), which identifies Pearson correlation as a reliable metric. Pearson preserves activation magnitude, is robust to imbalance, and aligns with the linear structure of SAE latents. Although the labels $y$ are discrete, SAE activations $x$ are continuous, making correlation an appropriate measure for selecting task-relevant features.
>
> ## 4. Interpretability as primary contribution
>
> **Reviewer Concern**: Relation to monosemanticity, concept benchmarks, and probe-based baselines.
>
> **Author Response:**
> CorrSteer relies on the SAE’s sparse latent decomposition but does not depend on feature interpretability. Instead, it performs task-specific optimization using correctness signals from generated tokens. Therefore, we evaluate CorrSteer on standard performance benchmarks, not AxBench-style concept evaluations.
>
> Feature descriptions (Neuronpedia) are used only as post-hoc diagnostics to confirm task alignment, not as part of the method. Because CorrSteer does not perform supervised feature prediction, linear probes are not an appropriate baseline, and we clarified this distinction in the revision.
>
> Following your suggestions, we moved the relevant interpretability-related analyses (cross-task transferability, pooling ablation, and negative-feature ablation) from the appendix into the main text, while placing Algorithm 1 in the appendix.

---

> ### Author Response · Authors · 2025-11-27
> **Official Comment by Authors**
>
> Dear Reviewer,
>
> We have addressed all your concerns with clarifications and additional analysis:
> - Clarified CorrSteer as an inference-time steering technique (not PEFT)
> - Provided formal variant definitions (Appendix A.2) to clarify the three ablations
> - Added multi-seed statistics and baseline SER results (Table 4)
> - Justified Pearson correlation choice following NeuralEval framework
>
> We would appreciate your feedback on our response. If there are any remaining questions, we are happy to provide further clarification. Otherwise, we would be grateful if you would consider updating your evaluation given these improvements.

---

### Author Response · Authors · 2025-12-03
**Summary of Updates**

We thank all reviewers for their constructive feedback. The revision addresses the main concerns as follows:

## 1. Baseline Comparisons and Statistical Robustness

We added adapted contrastive baselines (CAA, SPARE, DSG) using the same generation-time SAE activations and datasets. Table 4 now reports all methods with multi-seed statistics and Side Effect Ratio (SER) metrics. CorrSteer differs from contrastive methods by using correlation-based feature selection rather than activation differences, which can become unstable in sparse SAE space.

## 2. Technical Clarity and Formal Definitions

We added **Appendix A.2** with formal variant definitions explicitly connecting correlation $r_i^\ell$ (feature selection) to steering coefficient $c_i^\ell$ (magnitude):
- **CorrSteer-S**: $\mathcal{F_S} = \{\arg\max_{(\ell,i)} r_i^\ell : r_i^\ell > 0\}$ (single global feature)
- **CorrSteer-A**: $\mathcal{F_A} = \{(\ell, i) : i = \arg\max_j r_j^\ell, r_i^\ell > 0, \forall \ell\}$ (all layers)
- **CorrSteer-P**: $\mathcal{F_P} = \{(\ell, i) \in \mathcal{F_A} : \text{Acc}^{\text{val}}(\{(\ell,i)\}) > \text{Acc}^{\text{val}}(\emptyset)\}$ (validation-pruned)

Additionally:
- **Context handling**: Clarified that correlations operate at sequence level using pooled activations, capturing context through attention-mixed hidden states
- **Figure 1**: Improved caption explaining correlation between activation values and correctness

## 3. Presentation and Analysis

We moved key tables (5, 8) and interpretability analyses from appendix to main text, and moved Algorithm 1 to appendix. Added Appendix A.9 (SAE architecture differences) and Appendix A.10 (layer-wise correlation patterns).

---

**Summary:** The revision adds baseline comparisons with multi-seed statistics, technical definitions, and interpretability analyses. All missing citations have been added.

---

### Meta-Review · Area_Chair_CANE · 2025-12-30

**Summary:**

The paper presents CorrSteer, a method for steering LLMs using SAE features. Unlike previous methods that rely on contrastive datasets (positive vs. negative examples), CorrSteer identifies features based on their correlation with correctness during generation and automates the calculation of steering coefficients.

While reviewers initially praised the comprehensiveness of the evaluations and the interpretability of the results, primary concerns included
- A lack of comparison with established steering techniques like CAA, SPARE, or DSG
- Confusion regarding the relationship between correlation ($r$) and steering coefficients ($c$), and how context is handled
- Concerns about statistical significance and the "max@3" reporting style of the three variants
- Debate over whether the method should be framed as a performance booster or an interpretability tool

**Reviewer Concerns:**

Addressed concerns:
- The authors integrated adapted versions of CAA, SPARE, and DSG into their results. They clarified that while these methods can be adapted to the task, CorrSteer’s correlation-based selection is more stable in sparse activation spaces than simple activation differences
- The authors provided multi-seed statistics and expanded the reporting of the Side Effect Ratio (SER) for all baselines, reducing concerns about "max@3" overfitting

Oustanding concerns:
- Reviewer pRxr remained somewhat skeptical about the practical advantage of a positive-only regime, noting that for most benchmarks, negative/incorrect data is just as easy to acquire as positive data
- While the method matches fine-tuning in some cases, Reviewer igPZ’s point remains that as a performance-booster, it still lags behind more complex PEFT methods, suggesting the paper's true value may still lie more in its interpretability insights than as a state-of-the-art accuracy improver
- Although authors provide explanations regarding the correlation $r_i$ and steering coefficient $c_i$ and clarification about figure1, it still confusing to correctly understand the real meaning behind those content

**Reviewer Scores:**

Based on the addressed concerns, the addition of multi-seed statistics could solve the reviewer igPZ's particial concern. However, the remaining concerns from the outstanding part is still unsolved. In addition, although authors added more baseliens following reviewer pRxr's comments, it is still unclear whether those comparison is fair or not based on the positive-only regime used in this work. For reviewer kbSg, the response regarding context-handling and the formal link between $r_i$ and $c_i$ and figure 1 explanations is still unclear to fully understand the real meaning behind.

---

### Decision · Program_Chairs · 2026-01-26

Reject